# High-intensity interval training remodels the proteome and acetylome of human skeletal muscle

Morten Hostrup[1]*[†], Anders Krogh Lemminger[1][†], Ben Stocks[2][†], Alba Gonzalez-Franquesa[2], Jeppe Kjærgaard Larsen[2], Julia Prats Quesada[2], Martin Thomassen[1], Brian Tate Weinert[3], Jens Bangsbo[1], Atul Shahaji Deshmukh[2,4]*

[1]Section of Integrative Physiology, Department of Nutrition, Exercise and Sports, University of Copenhagen, Copenhagen, Denmark; [2]Novo Nordisk Foundation Center for Basic Metabolic Research, Faculty of Health and Medical Sciences, University of Copenhagen, Copenhagen, Denmark; [3]The Novo Nordisk Foundation Center for Biosustainability, Technical University of Denmark, Copenhagen, Denmark; [4]The Novo Nordisk Foundation Center for Protein Research, Clinical Proteomics, Faculty of Health and Medical Sciences, University of Copenhagen, Copenhagen, Denmark

*For correspondence:
mhostrup@nexs.ku.dk (MH);
atul.deshmukh@sund.ku.dk
(AShahajiD)

[†]These authors contributed equally to this work

Competing interest: The authors declare that no competing interests exist.

**Abstract** Exercise is an effective strategy in the prevention and treatment of metabolic diseases. Alterations in the skeletal muscle proteome, including post-translational modifications, regulate its metabolic adaptations to exercise. Here, we examined the effect of high-intensity interval training (HIIT) on the proteome and acetylome of human skeletal muscle, revealing the response of 3168 proteins and 1263 lysine acetyl-sites on 464 acetylated proteins. We identified global protein adaptations to exercise training involved in metabolism, excitation-contraction coupling, and myofibrillar calcium sensitivity. Furthermore, HIIT increased the acetylation of mitochondrial proteins, particularly those of complex V. We also highlight the regulation of exercise-responsive histone acetyl-sites. These data demonstrate the plasticity of the skeletal muscle proteome and acetylome, providing insight into the regulation of contractile, metabolic and transcriptional processes within skeletal muscle. Herein, we provide a substantial hypothesis-generating resource to stimulate further mechanistic research investigating how exercise improves metabolic health.

## Editor's evaluation

The authors have measured the proteome and acetylome of human skeletal muscle before and after high intensity exercise. They found that many of the subunits of Complex V in mitochondria are selectively acetylated after exercise. This study will serve as a very useful resource for people interested in muscle and the acetyl proteome.

## Introduction

Exercise training is the most effective means to improve cardiovascular fitness and metabolic health, both being predominant determinants of life expectancy and all-cause mortality (*Mandsager et al., 2018*). Alterations to the proteome of skeletal muscle underpin its adaptations to exercise training, including glucose and fatty acid metabolism, insulin sensitivity, mitochondrial respiration, immune function, and excitation-contraction coupling (*Deshmukh et al., 2021*; *Egan and Zierath, 2013*;

*Gonzalez-Franquesa et al., 2021*; *Holloway et al., 2009*; *Hostrup and Bangsbo, 2017*; *Robinson et al., 2017*; *Ubaida-Mohien et al., 2019*). Furthermore, proteome-wide post-translational modifications play an important role in regulating metabolism via modulating signaling, protein stability and enzyme activity (*Baeza et al., 2016*; *Cohen, 2000*), and are sensitive to exercise stimuli (*Hoffman et al., 2015*; *Overmyer et al., 2015*; *Parker et al., 2020*).

The application of mass-spectrometry-based proteomics has vastly expanded the catalog of post-translationally modified proteins, and in doing so has revealed novel insights into both histone and non-histone acetylation (*Choudhary et al., 2014*; *Hansen et al., 2019*; *Narita et al., 2019*; *Svinkina et al., 2015*). Nevertheless, the understanding of the acetylome in human skeletal muscle is still incomplete and remains to be studied in response to exercise training. Numerous cellular processes are regulated by protein acetylation, including transcription, metabolism, apoptosis, growth and muscle contraction among others (*Narita et al., 2019*). Lysine acetylation is an evolutionarily conserved post-translational modification, whereby lysine acetyltransferases catalyze the transfer of an acetyl group from acetyl-CoA to the ε-amino acid side chain of lysine and deacetylases remove acetyl groups from lysine residues. Alternatively, acetyl groups can be non-enzymatically transferred to lysine from acetyl-CoA (*Hansen et al., 2019*; *James et al., 2018*; *Pougovkina et al., 2014*; *Weinert et al., 2015*). In human *vastus lateralis* muscle biopsies from male athletes, 941 acetylated proteins containing 2811 lysine acetylation sites have been identified (*Lundby et al., 2012*). Mitochondria were the cellular component of human skeletal muscle exhibiting the greatest proportion of acetylated proteins (*Lundby et al., 2012*). Furthermore, the sensitivity of the acetylome to physiological stimuli has been demonstrated in rodents in response to acute fasting and chronic caloric restriction in liver (*Hebert et al., 2013*; *Pougovkina et al., 2014*; *Weinert et al., 2015*) as well as acute exercise (*Overmyer et al., 2015*) and high-fat diet (*Williams et al., 2020*) in skeletal muscle (*Overmyer et al., 2015*).

Continual advances in mass spectrometry-based proteomic technologies and approaches are increasing the depth and coverage of proteomic analyses. Nevertheless, proteome and proteome-wide post-translational modification analyses of skeletal muscle tissue remain highly challenging due to a wide dynamic range of protein abundances (*Deshmukh et al., 2015*). Single-shot data-dependent acquisition (DDA) analyses are often limited in their protein identifications and are characterized by relatively poor data completeness (i.e. high number of missing values; *Tabb et al., 2010*). However, the relatively recent development of data-independent acquisition (DIA) is helping to vastly reduce the proportion of missing data while simultaneously improving proteome depth in single-shot analyses of complex tissues (*Ludwig et al., 2018*). Nonetheless, DDA in combination with peptide-level fractionation remains a useful approach to overcome sample complexity, particularly when performing post-translational modification proteomics.

In this study, untrained men undertook 5 weeks of high-intensity interval training (HIIT) and *vastus lateralis* muscle biopsies were collected before and after HIIT for proteomic and lysine-acetylome analyses. We investigated the adaptation to HIIT in a deep human skeletal muscle proteome and undertook the first investigation of the acetylomic adaptations to exercise training within human skeletal muscle. HIIT remodeled skeletal muscle favoring mitochondrial biogenesis, but induced changes in protein abundance indicative of reduced calcium sensitivity. Furthermore, HIIT increased acetylation, particularly of the mitochondria and enzymes of the tricarboxylic acid (TCA) cycle.

## Results and discussion
### Physiological adaptations to HIIT

Eight untrained men (23–38 years of age; *Figure 1—figure supplement 1*) completed a HIIT regimen (*Figure 1A*) that consisted of 5 weeks of supervised cycling, performed as 4–5×4 min intervals at a target heart rate of >90% max interspersed by 2 min of active recovery, undertaken three times weekly. Participants were healthy, non-smokers and occasionally active, but otherwise untrained. Characteristics of the participants are presented in *Figure 1B*. Participants experienced a 14% and 17% improvement in maximal oxygen consumption ($VO_{2max}$) and incremental peak power output, respectively, during HIIT (*Figure 1B and C*; p<0.001), which compares favorably to typical HIIT-induced adaptations (*Wen et al., 2019*). In addition, participants gained an average of 1.0±0.3 kg lean mass during HIIT training, without changes in fat mass (*Figure 1B*). The rate of fat oxidation during submaximal

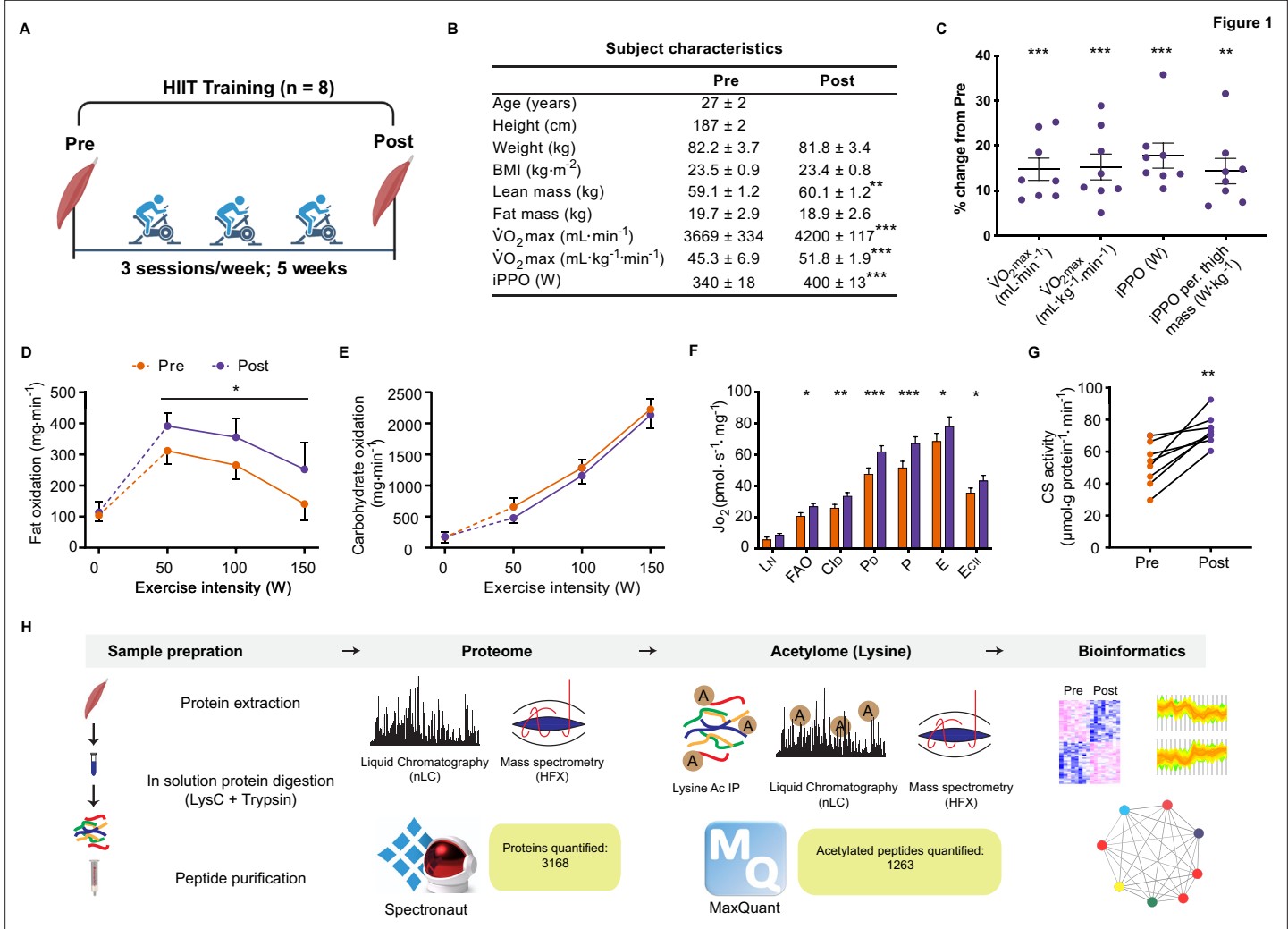

**Figure 1.** Physiological adaptations to HIIT. (**A**) Experimental overview. (**B**). Subject characteristics. (**C**). Five weeks of HIIT increased maximal oxygen uptake (VO₂max) and incremental peak power output (iPPO). (**D**) HIIT increased whole-body fat oxidation during submaximal exercise (50–150 W) without altering (**E**). carbohydrate oxidation. (**F**). HIIT increased mitochondrial respiration in skeletal muscle ($L_N$: leak respiration, FAO: fatty acid oxidation, $CI_D$: submaximal CI respiration, $P_D$: submaximal CI +II respiration, P: oxidative phosphorylation capacity, E: electron transport system capacity, $E_{CII}$: succinate-supported electron transport system capacity). (**G**). HIIT increased skeletal muscle citrate synthase (CS) activity. (**H**). Analytical workflow. Summary statistics are mean ± SEM (n=8). * $p<0.05$, ** $p<0.01$, *** $p<0.001$.

The online version of this article includes the following source data and figure supplement(s) for figure 1:

**Source data 1.** Source data for *Figure 1*.

**Figure supplement 1.** CONSORT flow diagram of enrolled participants.

**Figure supplement 2.** Proteome quality control.

cycling (50–150 W) increased following HIIT (*Figure 1D*), while no apparent changes were observed for carbohydrate oxidation (*Figure 1E*).

## HIIT increases mitochondrial respiratory capacity of skeletal muscle

Resting muscle biopsies were collected from the *vastus lateralis* before and three days after the final training session. Mitochondrial respirometry analyses were immediately performed on the freshly excised muscle samples using a substrate-uncoupler-inhibitor titration protocol (*Pesta et al., 2011*) on a high-resolution O2K respirometer (Oroboros, Innsbruck, Austria). In line with previous findings (*Christensen et al., 2016*; *Dohlmann et al., 2018*; *Granata et al., 2016*; *Jacobs et al., 2013*; *Larsen et al., 2015*; *Meinild Lundby et al., 2018*), HIIT increased mitochondrial respiration across a range

of respiratory states, increasing CI +II coupled respiration by 30% (p<0.001) and electron transport system capacity by 15% (p=0.049) (*Figure 1F*). Furthermore, HIIT induced a robust increase in citrate synthase activity by 42% (p=0.002) (*Figure 1G*), a validated marker of mitochondrial volume in human skeletal muscle (*Larsen et al., 2012*).

## Elevated mitochondrial respiration is underpinned by increased abundance of mitochondrial proteins following HIIT

Skeletal muscle biopsy samples were prepared for liquid-chromatography tandem mass spectrometry and measured using single-shot DIA (*Ludwig et al., 2018*; *Figure 1H*). We obtained extensive proteome coverage identifying 3343 proteins (*Supplementary file 1*). After applying a filter of four valid values out of the eight participants in at least one time point, 3168 proteins were quantified within skeletal muscle (*Supplementary file 2*). Due to the nature of DIA, few missing values required imputation in each sample (*Figure 1—figure supplement 2A*; mean proteins quantified per sample: 2853±33). Good correlation (median correlation: 0.88) between biological replicates (participants at the same time point) was apparent (*Figure 1—figure supplement 2B*). Furthermore, the proteome contained good coverage across multiple cellular compartments (*Figure 1—figure supplement 2C*). Principal component analysis demonstrated inter-sample variation; however, samples did cluster by time point (*Figure 1—figure supplement 2D*). Overall, our DIA proteome analysis displayed extensive coverage, demonstrated by the quantification of low abundant proteins such as transcription factors (e.g. CREB1 and TFEB) and myokines (e.g. IL18; *Supplementary file 1*).

HIIT regulated 126 proteins (permutation-based FDR <0.05). Of these, 102 proteins were upregulated, while 24 were downregulated (*Figure 2A*, *Supplementary file 2*). These included the upregulation of classical endurance exercise training-responsive mitochondrial proteins (e.g. cytochrome c oxidase subunit 5b; COX5B & NADH dehydrogenase 1 alpha subcomplex subunit 7; NDUFA7), as well as proteins involved in $NAD^+$ metabolism (e.g. nicotinamide phosphoribosyltransferase; NAMPT), branched-chain amino acid metabolism (branched-chain alpha-ketoacid dehydrogenase kinase; BCKDK) and ubiquinone biosynthesis (5-demethoxyubiquinone hydroxylase; COQ7), the latter of which we have recently described in exercise-trained mouse skeletal muscle (*Gonzalez-Franquesa et al., 2021*; *Figure 2A*). We also identified novel exercise-training regulated proteins, including, but not limited to, glutaminyl-tRNA synthase (QARS) and rab GDP dissociation inhibitor alpha (GDI1). QARS, which catalyzes the glutamate aminoacylation of tRNA and thus plays a central role in translation (*Deutscher, 1984*), was amongst the most significantly upregulated proteins within the proteome. In addition, GDI1, which inhibits insulin-stimulated glucose uptake by preventing the dissociation of GDP from Rab10 (*Chen et al., 2009*), was lowered, providing a novel potential mechanism for the insulin-sensitizing effects of exercise training.

After filtering for differentially regulated proteins, enrichment analysis using Fisher's exact test indicated the predominant regulation of mitochondrial proteins (e.g. GOCC: mitochondrial part and mitochondrial inner membrane) (*Figure 2B*, *Supplementary file 3*). Hierarchical clustering analysis on z-scored differentially regulated proteins identified that the mitochondrial terms were enriched within the upregulated proteins (*Figure 2C*). Summed protein abundances also demonstrated mitochondrial biogenesis (*Figure 2D*), including upregulated protein content of electron transport chain complexes (*Figure 2E* and *Figure 2—figure supplement 1*), mitochondrially encoded proteins (*Figure 2F*) and proteins containing a mitochondria-targeting transit peptide (*Figure 2G*). Thus, the increase in the proteome mirrors the functional increase in mitochondrial respiratory capacity of skeletal muscle after HIIT (*Figure 1F*).

## HIIT regulates proteins involved in skeletal muscle excitation-contraction coupling

We next investigated whether HIIT influenced the fiber-type proportions of skeletal muscle. While HIIT did not change skeletal muscle fiber-type composition in terms of myosin heavy chain isoforms (*Figure 3A*) and myosin light chain proteins (*Figure 3B*), a coordinated regulation of proteins modulating myosin light chain phosphorylation was identified (*Figure 3C*). Abundance of myosin light chain kinase 2 (MYLK2) decreased concomitantly with a reduction in smoothelin-like protein 1 (SMTNL1), an inhibitor of the myosin phosphatase complex (*Wooldridge et al., 2008*) and an increase in myosin phosphatase-targeting subunit 1 (PPP1R12A), the myosin-targeting regulatory subunit of protein

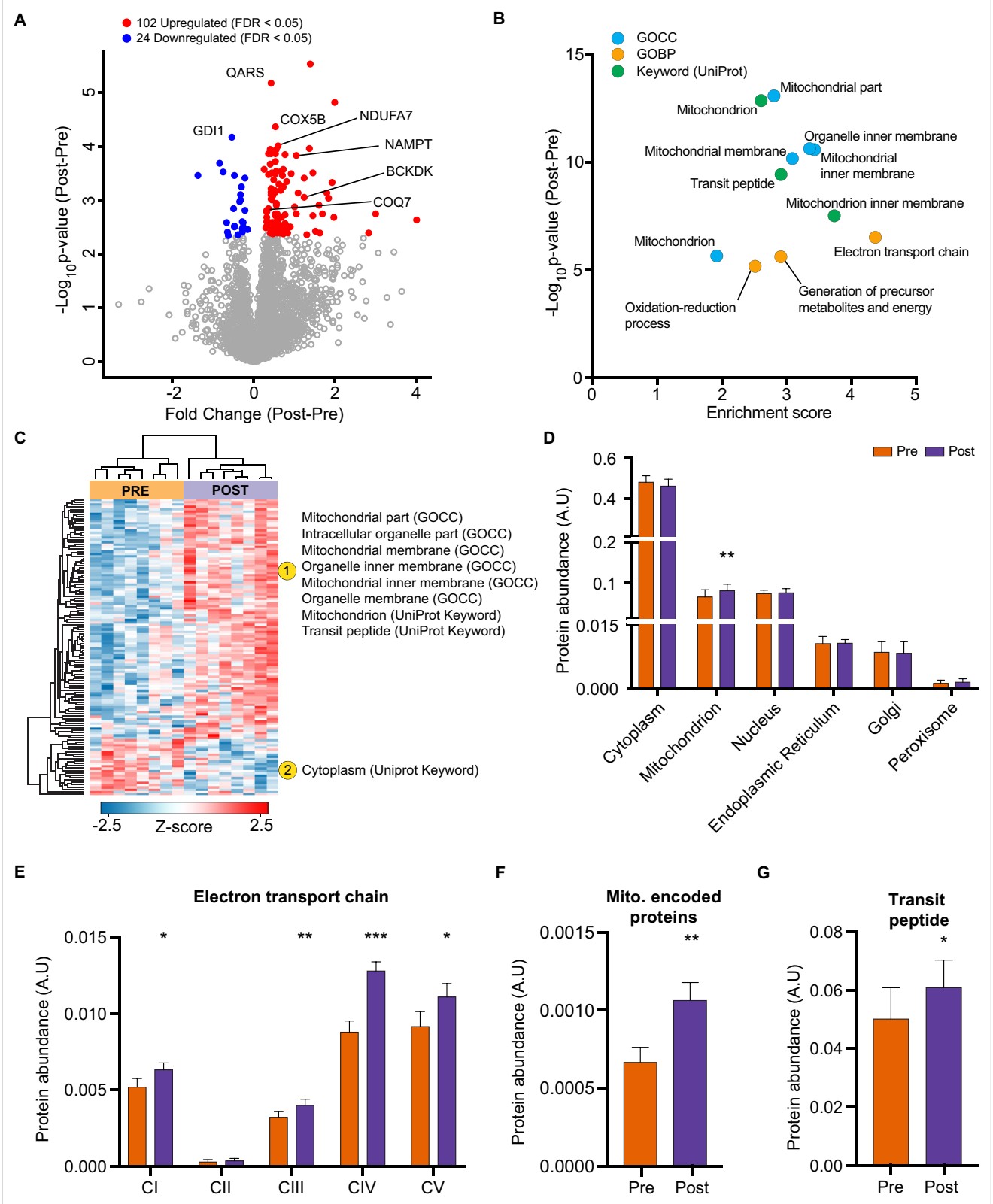

**Figure 2.** HIIT increases mitochondrial proteins and reduces a subset of contractile fiber associated proteins. (**A**) Volcano plot displaying 102 upregulated and 24 downregulated proteins following HIIT (FDR <0.05). (**B**). Fisher's exact tests identified the enrichment of mitochondrial terms within the differentially regulated proteins (enrichment analysis FDR <0.02). (**C**). Hierarchical clustering and enrichment analysis (enrichment analysis FDR <0.02) on differentially regulated proteins identified that mitochondrial terms are enriched within the upregulated proteins, while cytoplasm (UniProt

*Figure 2 continued on next page*

*Figure 2 continued*

Keyword) is enriched amongst the downregulated proteins. (**D**). Summed total protein abundances for different organelles (UniProt Keyword) shows upregulation of the mitochondrial protein content. (**E–G**). Summed total protein abundances display upregulation of electron transport chain complexes (**E**), mitochondrially encoded proteins (**F**) and proteins with a transit peptide (**G**). Summary statistics are mean ± SEM (n=8). * p<0.05, ** p<0.01, *** p<0.001.

The online version of this article includes the following source data and figure supplement(s) for figure 2:

**Figure supplement 1.** HIIT increases electron transport chain subunits.

**Figure supplement 1—source data 1.** Full image and annotation of OXPHOS immunoblot.

**Figure supplement 1—source data 2.** Raw ImageLab file of OXPHOS immunoblot.

**Figure supplement 1—source data 3.** Raw quantification data of OXPHOS immunoblot.

phosphatase 1 (*Alessi et al., 1992*; *Figure 3C*). Reduced MYLK2 abundance was confirmed by immunoblotting (*Figure 3—figure supplement 1A*). Furthermore, we have previously detected these adaptations in slow- and fast-twitch muscle fibers following moderate-intensity continuous cycling training (*Deshmukh et al., 2021*). Given that phosphorylation of myosin light chains promotes myofibrillar calcium sensitivity (*Davis et al., 2001*), the reduction in both MYLK2 and SMTNL1 alongside an increase in PPP1R12A indicates an adaptation towards reduced myosin phosphorylation and lowered calcium sensitivity following exercise training. Indeed, myofibrillar calcium sensitivity decreases following high-intensity sprint training in untrained males, particularly at a low pH representing physiological exercising conditions (*Lynch et al., 1994*). Although the functional importance of such adaptations remains elusive, knockdown of Smtnl1 augments exercise training-induced improvements in endurance performance, while sedentary Smtnl1$^{-/-}$ mice display a fiber-type switch mimicking endurance exercise training (*Wooldridge et al., 2008*). While we did not observe any concomitant myosin fiber-type switching (*Figure 3A*), the changes observed in proteins regulating myosin phosphorylation following HIIT could result in a right-shift of the Ca$^{2+}$-force relationship, meaning that a higher myoplasmic calcium concentration would be required for a given force production. This decline in myofibrillar calcium sensitivity may, however, support faster off-kinetics of calcium from troponin-C to be re-sequestered in the sarcoplasmic reticulum. Therefore, the changes observed in proteins regulating myosin phosphorylation following HIIT provide an intriguing mechanism for a HIIT-induced reduction in calcium sensitivity and warrants further attention.

Providing a further link to the altered calcium handling following HIIT (*Lynch et al., 1994*), we identified a downregulation of L-type calcium channel subunits, otherwise known as the dihydropyridine receptor (DHPR) (*Figure 3D and E*). DHPR controls the coupling of membrane depolarization and sarcoplasmic reticulum Ca$^{2+}$-release in skeletal muscle via its interaction with ryanodine receptor 1 (RYR1) (*Flucher and Franzini-Armstrong, 1996*). Conversely to DHPR, HIIT did not influence the abundance of RYR1 (*Figure 3F*) or the sarcoplasmic/endoplasmic reticulum calcium ATPases 1–3 (ATP2A1-3 (SERCAs); *Figure 3—figure supplement 1B*). The observation that DHPR abundance declined following HIIT points to a 'slowing' of muscle fibers. Indeed, a period of training can induce rapid changes in the abundance of proteins with importance for calcium handling, irrespective of changes in fiber type distribution (*Hostrup et al., 2019*; *Hostrup et al., 2018*; *Majerczak et al., 2008*; *Munkvik et al., 2010*; *Ortenblad et al., 2000*). Fast-twitch fibers have greater sarcoplasmic reticulum volume, faster calcium release and re-uptake kinetics, and a greater content of DHPR than slow-twitch fibers (*Banas et al., 2011*; *Deshmukh et al., 2021*; *Murgia et al., 2017*). Thus, the HIIT-induced reduction of DHPR abundance may represent a slowing of muscle fibers independently of their myosin heavy chain abundance.

## Lysine acetylomics of skeletal muscle identifies the predominant acetylation of mitochondrial and acetyl-coA metabolic proteins

Skeletal muscle biopsy samples were prepared for lysine acetylomics using a PTMscan Acetyl-Lysine Motif kit (Cell Signaling Technology) (*Svinkina et al., 2015*) and measured via liquid-chromatography tandem mass spectrometry using DDA. Due to limited sample availability from one participant, lysine acetylomics was performed for 7 of the participants. We identified a total of 1990 acetyl-sites on 664 proteins (*Supplementary file 4*). While this is fewer than the 2,11 acetyl-sites on 941 proteins identified in the largest human skeletal muscle acetylome to date (*Lundby et al., 2012*), we identified 1073

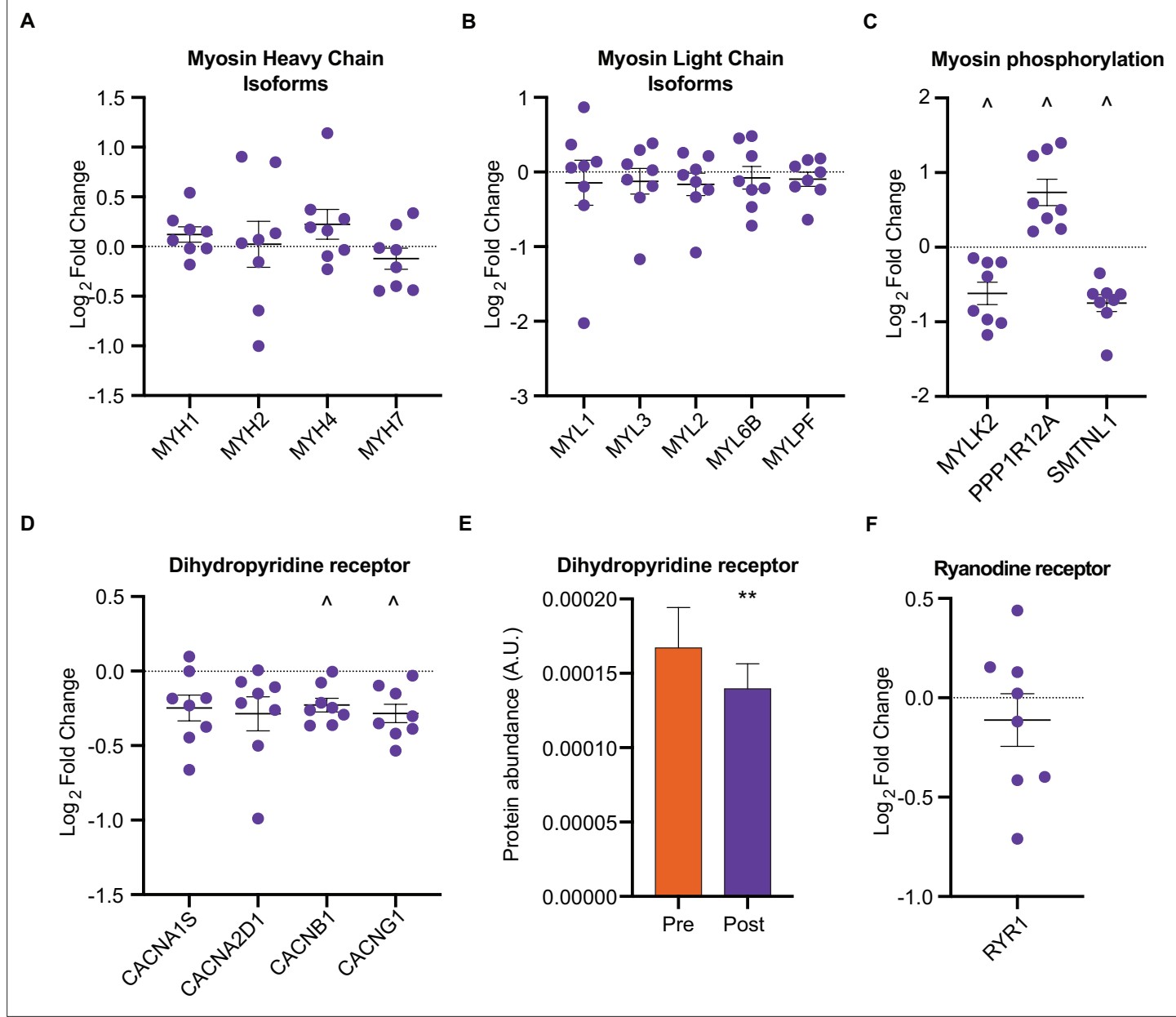

**Figure 3.** HIIT regulates proteins involved in skeletal muscle calcium sensitivity and handling HIIT did not alter the abundance of myosin heavy chain (MYH1: MyHC2x, MYH2: MyHC2a, MYH4: MyHC2x, MYH7: MyHCβ; **A**) or light chain (**B**) isoforms. (**C**). HIIT regulates proteins controlling myosin phosphorylation. (**D**). HIIT reduces abundance of subunits of the dihydropyridine receptor. (**E**). Summed total protein abundances display downregulation of the dihydropyridine receptor. (**F**). HIIT does not alter the abundance of ryanodine receptor 1. Summary statistics are mean ± SEM (n=8). ^ FDR <0.05. * p<0.05, ** p<0.01, *** p<0.001.

The online version of this article includes the following source data and figure supplement(s) for figure 3:

**Figure supplement 1.** HIIT decreases MYLK2 abundance.

**Figure supplement 1—source data 1.** Full image and annotation of MYLK2 immunoblot.

**Figure supplement 1—source data 2.** Raw ImageLab file of MYLK2 immunoblot.

**Figure supplement 1—source data 3.** Raw quantification data of MYLK2 immunoblot.

acetyl-sites that were not identified in the analysis of Lundby et al (*Figure 4—figure supplement 1A*). The difference in acetyl-sites identified between the two studies likely reflects a combination of exercise-induced acetylation revealing novel-acetyl sites, and the stochastic nature of data-dependent acquisition, the different LC-MS/MS instruments employed, and the use of updated FASTA files.

Nonetheless, we substantially extend the human skeletal muscle acetylome. After applying a filter of 4 valid values out of the seven participants in at least one time point we quantified 1263 acetylated sites on 464 proteins within skeletal muscle (*Figures 1G and 4A*, *Supplementary file 5*). Six and 42 acetyl-sites were quantified exclusively in the pre- and post-HIIT samples, respectively. The mean number of acetyl-sites quantified in each sample was 976 (*Figure 4—figure supplement 1B*). Median correlation between biological replicates (between participants at the same time point) was 0.76 (*Figure 4—figure supplement 1C*), indicating greater variation in the human acetylome than the proteome, as would be expected (*Lundby et al., 2012*). The acetylome also displayed extensive coverage across multiple cellular compartments (*Figure 4—figure supplement 1D*), which largely reflected the abundance distribution of the proteome (*Figure 1—figure supplement 2C*), albeit with a slight relative increase in the mitochondrial identifications. Principal component analysis identified variation in the acetylome between samples (*Figure 4—figure supplement 1E*). Although there was no clear separation of samples by time point, a leftward shift is apparent on component 1 from pre to post samples for almost all participants (six out of seven). There was considerable overlap with proteins quantified in the proteome, with 421 proteins quantified in both (*Figure 4A*). Of the quantified acetylated sites (1263), the majority (1232) were located on these 421 proteins (*Figure 4A*). Single acetyl-sites were quantified on the majority of proteins, with a trend for decreasing frequency with additional acetyl-sites (*Figure 4B*). However, several proteins were quantified with a large number of acetylation sites (e.g. ≥15 acetyl-sites) (*Figure 4B*), examples of which include the contractile proteins titin (TTN) and slow-twitch myosin beta (MYH7).

After summing the median intensities of acetyl peptides from each protein in the pre-HIIT samples, we ranked the abundance of acetylated proteins within skeletal muscle (*Supplementary file 6*) and highlighted the top 10 proteins with the highest acetyl-intensity (*Figure 4C*). Histone H4 (HIST1H4A) had the highest acetylation intensity, making up approximately 10% of the total acetylome intensity, supporting the regulatory role of histone acetylation in modulating transcription (*Sterner and Berger, 2000*; *Strahl and Allis, 2000*). The majority of the remaining top acetylated proteins were metabolic enzymes, including the TCA cycle proteins malate dehydrogenase (MDH2) and fumarate hydratase (FH), as well as subunits of complex V (ATP5H and ATP5O) within the electron transport chain. Superoxide dismutase (SOD2), a canonical acetylated enzyme involved in mitochondrial reactive oxygen species handling (*St Clair et al., 1992*; *Tao et al., 2009*), also displayed high acetylation intensity. We further compared the protein abundance rank for these high-intensity acetylated proteins within the proteome (*Figure 4D*). These proteins did not appear to be the highest intensity acetylated proteins simply due to their relative protein abundance. While they all fell within the top 318 proteins in the proteome, they did not make up the most abundant proteins, except for creatine kinase M-type (CKM), which is extremely abundant in skeletal muscle (*Figure 4D*). To identify systematic trends in protein acetylation, we performed a one-dimensional enrichment analysis (*Cox and Mann, 2012*) on the summed protein acetylation intensities, which ranked the proteins by acetylation intensity and identifies Gene Ontology (GO) annotations that are systematically over-represented with higher intensities (positive enrichment factor) and lower intensities (negative enrichment factor). This identified mitochondrial terms (e.g. GOCC: mitochondrial part and mitochondrial matrix) as enriched for high acetylation intensity (*Figure 4E* and *Supplementary file 7*). In particular, proteins of carboxylic acid metabolic processes (e.g. TCA cycle) and monovalent inorganic cation transport (e.g. electron transport chain complex V proteins) display systematically high acetylation intensity (*Figure 4E* and *Supplementary file 7*). Furthermore, the Uniprot keyword term 'muscle protein' was also enriched for high acetylation intensity, a term that mainly encompasses the contractile machinery of skeletal muscle (e.g. myosins).

In order to examine acetylation stoichiometry within skeletal muscle, we estimated the relative stoichiometries of acetyl-sites using abundance-corrected intensities (ACI) (*Supplementary file 8*). ACIs were calculated by dividing acetyl-peptide intensities by the intensities of their corresponding protein, with ACI values correlating with stoichiometry (*Hansen et al., 2019*). After ranking acetyl-site ACI for the pre-HIIT samples, we highlighted the top 10 acetyl-sites with the highest ACI (*Figure 4F*). Trifunctional enzyme subunit alpha (HADHA), FH, HIST1H4A and 3- ketoacyl-CoA thiolase (ACAA2), which were identified as among the highest intensity acetylated proteins (*Figure 4C*), all contain high stoichiometry acetyl-sites (*Figure 4F*). However, nicotinamide nucleotide transhydrogenase (NNT) K70 had the highest ACI within skeletal muscle (*Figure 4F*). NNT is an inner mitochondrial membrane

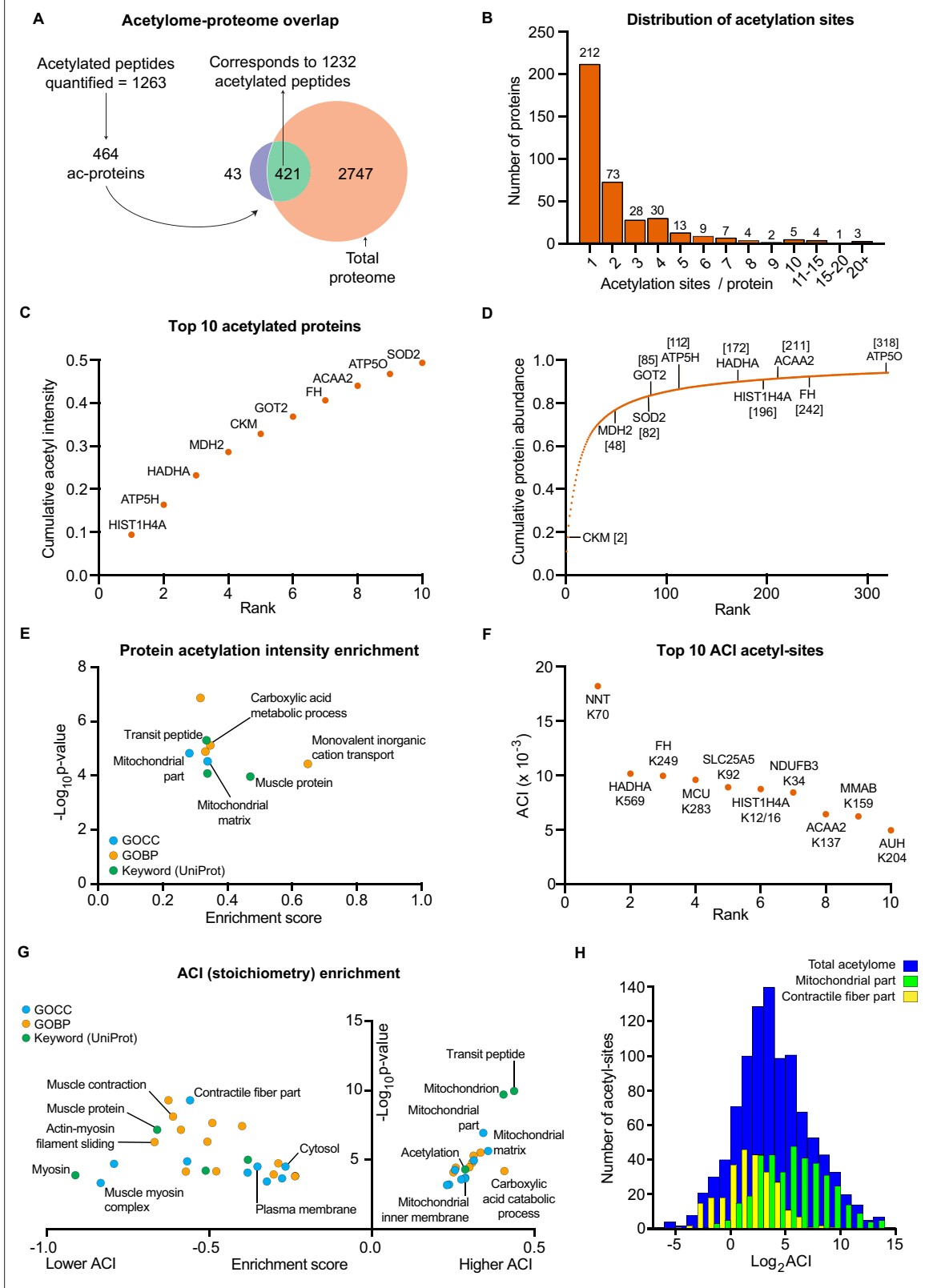

**Figure 4.** The human skeletal muscle acetylome displays higher stoichiometry on mitochondrial proteins and lower stoichiometry on contractile proteins. (**A**) We quantified 1263 acetyl-sites corresponding to 464 proteins (n=7). Of these, 421 proteins were also quantified in the proteome with 1232 of the quantified acetyl-sites located on these 421 proteins. (**B**). Distribution of acetyl-sites on proteins within the pre-HIIT acetylome. (**C**). Top 10 highest intensity acetylated proteins (acetyl-peptide intensities were summed for each protein) within the pre-HIIT acetylome. (**D**). Cumulative protein

*Figure 4 continued on next page*

*Figure 4 continued*

abundance and rank within the proteome (the top 10 highest intensity acetyl-proteins are highlighted). (**E**). One-dimensional enrichment analysis of acetylated protein intensity identified mitochondrial proteins, particularly those involved in carboxylic acid metabolism and monovalent inorganic cation transport (e.g. complex V) as having systematically high acetylation intensities (enrichment analysis FDR <0.02). (**F**). Top 10 acetyl-sites with highest abundance-corrected intensities (ACIs). (**G**). One-dimensional enrichment analysis of acetyl-site ACIs identified mitochondrial and carboxylic acid catabolic proteins as higher stoichiometry (positive enrichment factor), while contractile fiber cytosolic and plasma membrane proteins were enriched as lower stoichiometry negative enrichment factor; (enrichment analysis FDR <0.02; enrichment performed on leading protein IDs). (**H**). Histogram depicting the ACI distribution of the total acetylome (blue), the mitochondrial acetyl-sites (green) and the contractile fiber acetyl-sites (yellow). Mitochondrial proteins were distributed at higher ACI values and contractile fiber proteins at lower ACI values than the total acetylome.

The online version of this article includes the following figure supplement(s) for figure 4:

**Figure supplement 1.** Acetylome quality control.

protein, which uses the proton motive force to maintain high levels of NADPH (*Hoek and Rydström, 1988*) and in doing so can regulate metabolism (*Freeman et al., 2006*; *Ho et al., 2017*). NNT is also highly acetylated in cardiac muscle (*Foster et al., 2013*), although the effect of acetylation on NNT activity remains to be determined. Nonetheless, NNT can regulate acetylation via NADPH-mediated regulation of histone deacetylase 1 (HDAC1) activity (*Ho et al., 2017*), thus acetylation of NNT may represent a feedback mechanism controlling cellular acetylation and metabolism.

To identify systematic trends in acetyl-site stoichiometry, we performed a one-dimensional enrichment analysis (*Cox and Mann, 2012*) on acetyl-site ACI using the leading protein ID for relative enrichment (*Tyanova et al., 2016*). We identified the mitochondria (e.g. GOCC: mitochondrial part and mitochondrial inner membrane) and carboxylic acid catabolic processes to be enriched within higher stoichiometry acetylated proteins (*Figure 4G, H* and *Supplementary file 9*). This extends previous analyses by not only showing that mitochondrial proteins make up the majority of acetylated proteins in skeletal muscle (*Lundby et al., 2012*), but also by identifying mitochondrial proteins as having relatively high stoichiometry acetyl-sites – an observation that is consistent with analyses in HeLa cells, rodent liver and yeast (*Hansen et al., 2019*; *Weinert et al., 2014*; *Weinert et al., 2015*). Enriched for proteins with low stoichiometry acetyl-sites were proteins involved in contraction (e.g. GOCC: contractile fiber part; *Figure 4G, H* and *Supplementary file 9*), including the Uniprot keyword term 'muscle protein' (*Figure 4G*), which, conversely, was enriched for high-protein acetylation intensity (*Figure 4E*). Myosin displayed the greatest enrichment for low acetylation stoichiometry (*Figure 4G*). Thus, despite contractile proteins displaying high-protein acetylation intensity and making up a substantial number of known acetylated proteins within skeletal muscle (*Lundby et al., 2012*), they contain acetylation sites of relatively low stoichiometry. The high acetylation intensity of these proteins is therefore likely due to high protein abundance and a large number of acetylation sites per protein. Proteins annotated to the cytosol and plasma membrane also displayed low acetyl-site ACI (*Figure 4G*). This highlights differences in the inter-organelle acetylation levels of skeletal muscle and can likely be explained by subcellular compartmentalization of acetyl-CoA (*Pietrocola et al., 2015*).

## HIIT induces acetylation of mitochondrial and TCA cycle proteins

HIIT increased the mean number of acetyl-sites quantified per sample by over 100 sites (Pre: 915±44, Post: 1037±51; p=0.015; *Figure 4—figure supplement 1B*). Using a stringent statistical approach of permutation-based FDR corrected paired t-tests, we identified 20 upregulated acetyl-sites and 1 downregulated site (FDR <0.05; *Figure 5A*). In addition, by applying a validated, albeit less stringent, significance score ($\Pi$-value), which combines the statistical significance (p-value) with the fold change (*Xiao et al., 2014*), we extended this to identify 257 upregulated and 26 downregulated acetyl-sites following HIIT ($\Pi$<0.05; *Figure 5A* and *Supplementary file 5*). In general, there was a trend for increased acetylation following HIIT (*Figure 5A*), which could not be explained by changes in protein content as the fold changes in acetylation generally exceeded those of protein abundance (*Figure 5B*). Therefore, we did not normalize the acetylome data to protein abundance. The increase in protein acetylation was subsequently validated via immunoblotting (*Figure 5—figure supplement 1A*). Within the increased sites, acetyl-sites on metabolic enzymes isocitrate dehydrogenase (IDH2), pyruvate dehydrogenase E1 component subunit alpha (PDH1A), succinate CoA ligase subunit alpha (SUCLG1) and ATP synthase coupling factor 6 (ATP5J) were apparent (*Figure 5A*), while hyperacetylation of

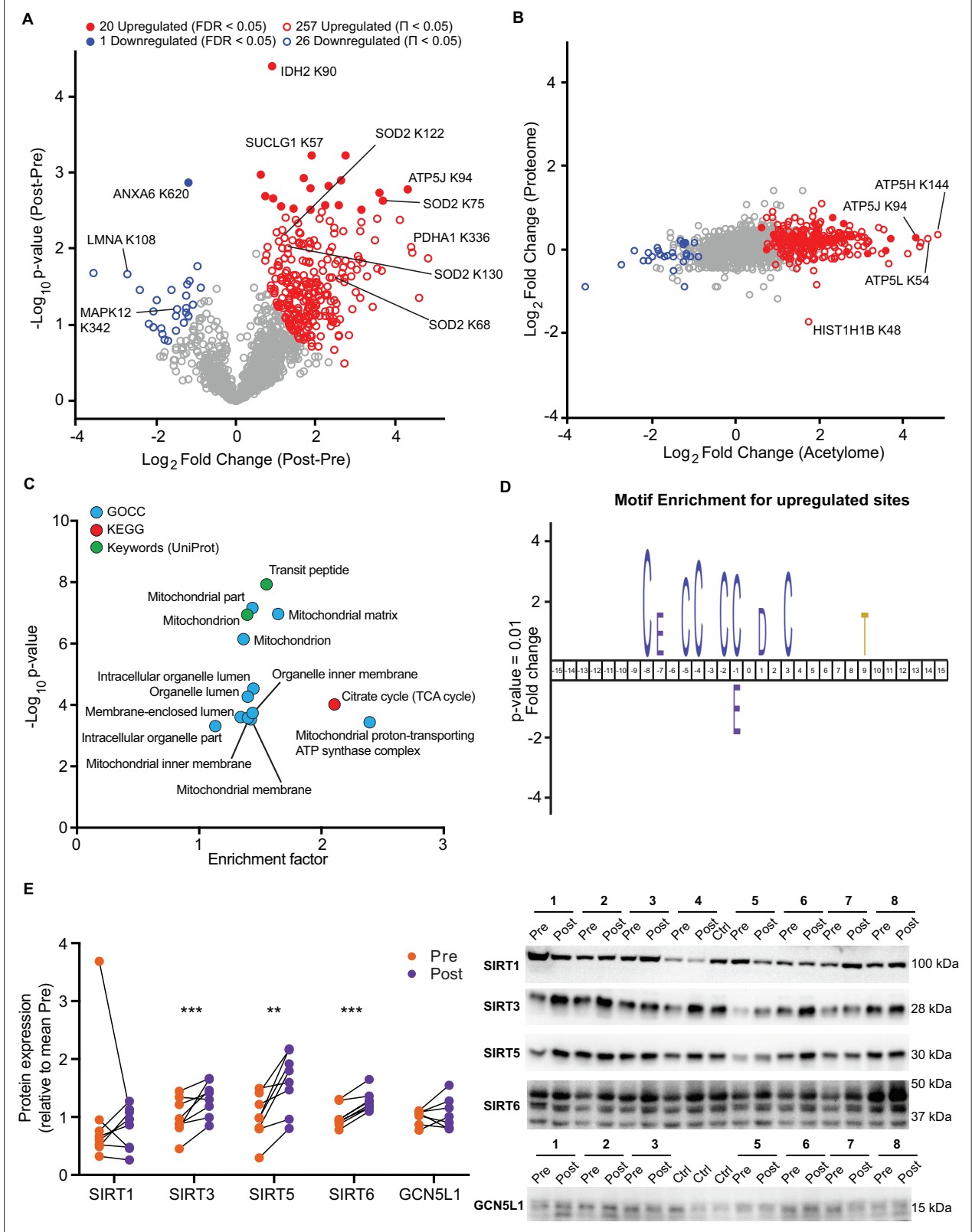

**Figure 5.** HIIT increases acetylation of mitochondrial and TCA cycle proteins concomitantly with an increase in SIRT3 abundance. (**A**) Volcano plot displaying 20 upregulated (filled red circles) and 1 downregulated (filled blue circle) acetyl-sites following HIIT at an FDR <0.05, while 257 acetyl-sites were upregulated (red circles) and 26 downregulated (blue circles) at ∏<0.05 (n=7). (**B**). Scatter plot indicating that HIIT-induced changes in acetyl-site intensity typically exceeded that of the corresponding protein. (**C**). Fisher's exact tests identified the enrichment of mitochondrial and TCA cycle terms

*Figure 5 continued on next page*

*Figure 5 continued*

within the differentially regulated acetyl-proteins (enrichment analysis FDR <0.02; enrichment performed on leading protein IDs). (**D**). IceLogo motif enrichment (p<0.01) for the upregulated sites displayed a predominance of proximal cysteine residues relative to the acetylated lysine (position 0). (**E**). Immunoblotting analysis identified the upregulation of the deacetylases SIRT3, SIRT5, and SIRT6, but no change in SIRT1, whilst the mitochondrial acetyltransferase GCN5L1 remained unchanged following HIIT (n=7–8). Representative images confirming equal loading are displayed in ***Figure 5— figure supplement 2***. * p<0.05, ** p<0.01, *** p<0.001.

The online version of this article includes the following source data and figure supplement(s) for figure 5:

**Source data 1.** Full image and annotation of SIRT1 immunoblot.

**Source data 2.** Raw ImageLab file of SIRT1 immunoblot.

**Source data 3.** Raw quantification data of SIRT1 immunoblot.

**Source data 4.** Full image and annotation of SIRT3 immunoblot.

**Source data 5.** Raw ImageLab file of SIRT3 immunoblot.

**Source data 6.** Raw quantification data of SIRT3 immunoblot.

**Source data 7.** Full image and annotation of SIRT5 immunoblot.

**Source data 8.** Raw ImageLab file of SIRT5 immunoblot.

**Source data 9.** Raw quantification data of SIRT5 immunoblot.

**Source data 10.** Full image and annotation of SIRT6 immunoblot.

**Source data 11.** Raw ImageLab file of SIRT6 immunoblot.

**Source data 12.** Raw quantification data of SIRT6 immunoblot.

**Source data 13.** Full image and annotation of GCN5L1 immunoblot.

**Source data 14.** Raw ImageLab file of GCN5L1 immunoblot.

**Source data 15.** Raw quantification data of GCN5L1 immunoblot.

**Figure supplement 1.** HIIT increases protein acetylation.

**Figure supplement 1—source data 1.** Full image and annotation of Pan-acetyl immunoblot.

**Figure supplement 1—source data 2.** Raw ImageLab file of Pan-acetyl immunoblot.

**Figure supplement 1—source data 3.** Raw quantification data of Pan-acetyl immunoblot.

**Figure supplement 1—source data 4.** Full image and annotation of ac-SOD2 K68 immunoblot.

**Figure supplement 1—source data 5.** Raw ImageLab file of ac-SOD2 K68 immunoblot.

**Figure supplement 1—source data 6.** Raw quantification data of ac-SOD2 K68 immunoblot.

**Figure supplement 1—source data 7.** Full image and annotation of ac-SOD2 K122 immunoblot.

**Figure supplement 1—source data 8.** Raw ImageLab file of ac-SOD2 K122 immunoblot.

**Figure supplement 1—source data 9.** Raw quantification data of ac-SOD2 K122 immunoblot.

**Figure supplement 2.** Representative images of equal loading for immunoblot analyses.

**Figure supplement 2—source data 1.** Full image and annotation of n=8 Coomassie stain.

**Figure supplement 2—source data 2.** Raw ImageLab file of n=8 Coomassie stain.

**Figure supplement 2—source data 3.** Full image and annotation of n=7 stain-free blot.

**Figure supplement 2—source data 4.** Raw ImageLab file of n=7 stain-free blot.

**Figure supplement 2—source data 5.** Full image and annotation of C2C12 stain-free blot.

**Figure supplement 2—source data 6.** Raw ImageLab file of C2C12 stain-free blot.

SOD2 occurred at K68, 75, 122, and 130 in response to HIIT (***Figure 5A***, ***Figure 5—figure supplement 1B***). Included within the downregulated sites were acetyl-sites on annexin A6 (ANXA6), laminin (LMNA) and p38 mitogen-activated protein kinase γ (MAPK12) (***Figure 5A***). Interestingly, Histone H1.5 (HIST1H1B) K48 showed increased acetylation, despite significantly reduced protein abundance (***Figure 5B***).

A Fisher's exact test of HIIT-regulated acetyl-sites (∏<0.05), using leading protein ID for relative enrichment, indicated the predominant regulation of the TCA cycle pathway, mitochondrial proteins (e.g. GOCC: mitochondrial part and mitochondrial inner membrane) and, in particular, electron transport chain complex V proteins (GOCC: mitochondrial proton-transporting ATP synthase complex) (***Figure 5C***, ***Supplementary file 10***). These GO terms were strikingly similar to the terms enriched

for high acetylation intensity and stoichiometry in the pre-HIIT samples (*Figure 4E and G*), indicating that proteins that are highly acetylated in the basal state are most susceptible to increased acetylation following HIIT. To further explore the regulation of HIIT-induced acetylation in skeletal muscle, we looked for consensus sequences around the acetylated lysine residues using iceLogo (*Colaert et al., 2009*). We identified a sequence with a predominance of proximal cysteine residues as well as an aspartic acid residue in the +1 position (*Figure 5D*). The aspartic acid at +1 is a feature of mitochondrial acetylated proteins (*Lundby et al., 2012*) and its enrichment is likely due to the over-representation of mitochondrial proteins in the upregulated acetyl-sites (*Figure 5C*). Proximal cysteine residues are associated with promoting non-enzymatic lysine acetylation (*Hansen et al., 2019*; *James et al., 2018*) via initial cysteine acetylation followed by transfer of the acetyl group to lysine (*James et al., 2018*). Indeed, non-enzymatic acetylation increases alongside elevated fatty acid oxidation (*Pougovkina et al., 2014*), as is apparent in skeletal muscle mitochondria after HIIT (*Figure 1E*). This likely occurs through changes in acetyl-CoA concentration and/or increased acetyl-CoA flux (*Pougovkina et al., 2014*). Although acetyl-CoA content within resting skeletal muscle does not change in response to exercise training (*Putman et al., 1998*), acetyl-CoA levels and TCA cycle flux increase during exercise (*Constantin-Teodosiu et al., 1991*; *Gibala et al., 1998*; *Howlett et al., 1998*; *Putman et al., 1998*), which may result in an accumulation of lysine acetylation during repeated bouts of exercise. Conversely, acute exercise decreases skeletal muscle acetylation in exercising rats (*Overmyer et al., 2015*), but whether this is the case for humans remains to be determined. Nonetheless, we identify a robust increase in skeletal muscle acetylation, predominantly of mitochondrial proteins, following 5 weeks of HIIT in humans.

Because HIIT altered the skeletal muscle acetylome, we examined proteins regulating acetylation in skeletal muscle. A subset of sirtuins (SIRT2, SIRT3, and SIRT5), a family of protein deacetylases, were quantified in the proteome. SIRT3, a mitochondrial-localized deacetylase, displayed a small but non-significant trend for increased abundance following exercise training (*Supplementary file 2*). To further investigate this, we immunoblotted for SIRT1, SIRT3, SIRT5, and SIRT6. Of the cystosolic and nuclear-localized SIRTs, the abundance of SIRT1 did not change, while SIRT6 increased by 27%. The mitochondrial-localized SIRT3 and SIRT5 increased in abundance by 61% and 32%, respectively, following HIIT (*Figure 5E*). Thus, mitochondrial acetylation increased concomitantly with an increase in the abundance of mitochondrial sirtuins (*Figure 5B and E*). Conversely, the protein abundance of the mitochondrial acetyltransferase general control of amino acid synthesis 5 like-1 (GCN5L1) was unchanged by exercise training (*Figure 5E*). Although the deacetylase activity of SIRT5 has been questioned, instead displaying a preference for desuccinylase and demalyonylase activity (*Du et al., 2011*; *Peng et al., 2011*), SIRT3 appears to play a major role in suppressing non-enzymatic mitochondrial acetylation (*Weinert et al., 2015*). Therefore, elevated SIRT3 abundance following exercise training might be interpreted as a mechanism to suppress excess acetylation, either to preserve the activity of mitochondrial enzymes (*Hirschey et al., 2010*; *Tao et al., 2010*; *Vassilopoulos et al., 2014*) or to scavenge acetyl groups for reintegration into acetyl-CoA and energy production.

Given the enrichment of mitochondrial terms and particularly of complex V proteins in the HIIT-regulated acetylated proteins, we filtered for proteins annotated to the different complexes of the electron transport chain using the HUGO database (*Braschi et al., 2019*) and highlighted the regulation of each acetyl-site on these proteins alongside the changes in protein abundance (*Figure 6A*). While proteins of complexes I, II, III and IV showed a mixed acetylation response to exercise training, almost every quantified complex V protein had at least one acetyl-site that increased following HIIT (*Figure 6A*). In fact, ATPase inhibitor (ATPIF1), one of the two complex V proteins that do not show elevated acetylation, is a negative regulator of complex V and is unlikely to be constitutively bound to the ATP synthase complex (*Campanella et al., 2008*). Why complex V displays elevated acetylation levels both in the basal (*Figure 4E*) and exercise trained states is unclear (*Figures 5C and 6A*). However, as a large proportion of ATP synthase extends into the mitochondrial matrix, many subunits are likely to be more exposed to acetyl-CoA than other membrane-embedded electron transport chain complexes. In support, we detected acetylation on all of the subunits of complex V that reside or extend into the mitochondrial matrix (ATP5A1, ATP5B, ATP5C1, ATP5D, ATP5E, ATP5F1, ATP5H, ATP5J, and ATP5O), while we only detect acetylation on two of the membrane-embedded subunits (ATP5L and MT-ATP8), despite detecting a further four membrane-embedded subunits in the proteome (ATP5I, ATP5J2, MT-ATP6, and USMG5). Indeed, the majority of acetylated proteins

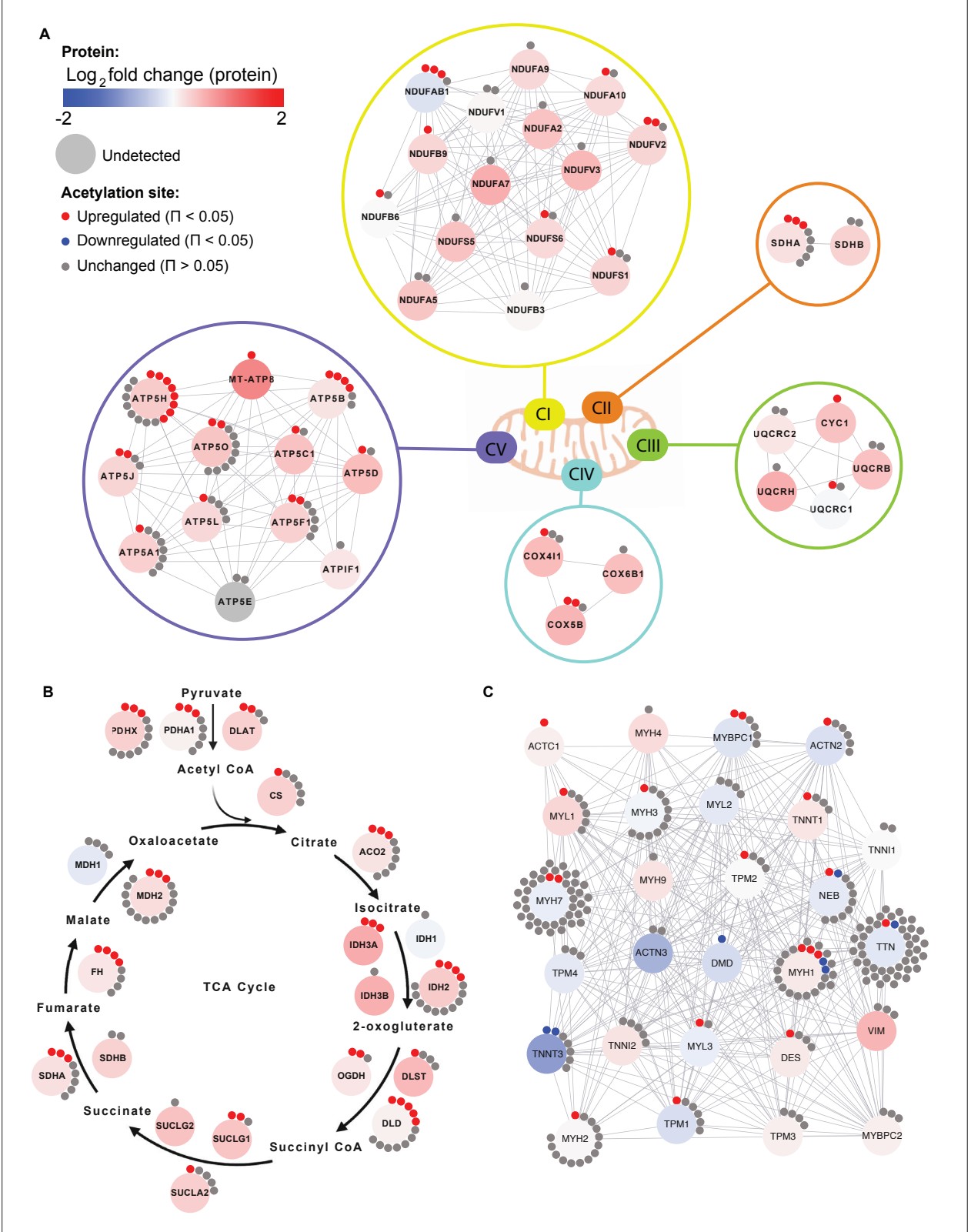

**Figure 6.** Individual acetyl-site regulation of mitochondrial, TCA cycle and contractile proteins following HIIT. Regulation of acetyl-sites on (**A**). electron transport chain complex subunits (annotated by HUGO), (**B**) TCA cycle proteins (annotated by KEGG) and (**C**) muscle contraction proteins (annotated by REACTOME).

quantified on each electron transport chain complex (CI-CV) are exposed to either the mitochondrial matrix or the intermembrane space. Similarly, TCA cycle enzymes, which are localized to the mitochondrial matrix, displayed high acetylation intensity and stoichiometry in the basal state and HIIT augmented levels of acetylation at every stage of the cycle (*Figure 6B*), again likely due to their proximity to acetyl-CoA. Conversely, cytosolic proteins regulating skeletal muscle contraction showed fewer upregulated acetyl-sites with a number of downregulated sites following HIIT (*Figure 6C*). Thus, proteins in close proximity to acetyl-CoA appear to be more susceptible to acetylation.

Despite widespread HIIT-induced acetylation of mitochondrial proteins, these data should be considered in the context of acetylation stoichiometry, which is generally low (*Weinert et al., 2015*). Median acetylation stoichiometry in rodent liver has been calculated at only 0.05%, which increases slightly to 0.11% for mitochondrial proteins (*Weinert et al., 2015*). Although acetylation is tissue-dependent (*Lundby et al., 2012*), skeletal muscle levels are unlikely to be vastly different. The median $\log_2$-fold change for mitochondrial acetyl-sites following HIIT was 0.9 (the corresponding value in the proteome was 0.3), indicating that median post-HIIT stoichiometry would have remained at less than 1%. Thus, the levels of HIIT-induced changes in acetylation may not be large enough to influence gross activity on most metabolic enzymes. Nonetheless, physiological acetylation may be high enough to be regulatory on some proteins. For example, acetylation of mitochondrial malate dehydrogenase (MDH2) negatively correlates with its enzyme activity (*Overmyer et al., 2015*). Another such enzyme could be pyruvate dehydrogenase (PDH), whose E1 component subunit alpha (PDHA1) is within the top 25 most acetylated proteins (*Supplementary file 6*), whilst PDHA1 Lys 336 was within the top 5 most upregulated acetyl-sites with exercise training (*Figure 5A*). Increased acetylation of PDHA1 occurs alongside suppressed enzyme activity in SIRT3 knockout mice (*Jing et al., 2013*). By catalyzing the reaction between pyruvate and acetyl-CoA, PDH provides the link between glycolysis and the tricarboxylic acid cycle (*Constantin-Teodosiu et al., 2012*). Thus, exercise-induced hyperacetylation of PDH could play a role in the metabolic switch towards elevated fat oxidation following exercise training. Furthermore, mitochondrial hyperacetylation increases fatty-acid-supported respiration (*Williams et al., 2020*). These data are consistent with the concomitant increase in acetylation (*Figure 5A*) and mitochondrial respiratory capacity and fat oxidation (*Figure 1F*) following HIIT, suggesting that hyperacetylation of mitochondrial proteins following exercise training may support the bioenergetic adaptations to exercise.

## HIIT increases histone acetylation

Aside from metabolic enzymes and contractile proteins, histones are regulated by acetylation (*Sterner and Berger, 2000*; *Strahl and Allis, 2000*) and may represent a more specifically regulated pool of proteins, owing to the nuclear localization of acetyltransferases (*Hendzel et al., 1994*). Increases in certain acetylated histone residues are well characterized to augment transcription, with exercise known to acutely regulate H3 acetylation (e.g. H3 K4, H3 K9/14, and H3 K36) (*Joseph et al., 2017*; *Lochmann et al., 2015*; *McGee et al., 2009*; *Smith et al., 2008*). Here, we identified 18 distinct acetylated peptides from histones, with acetylated peptides from H1.5 (HIST1H1B K48), H2B type 2 F (HIST2H2BF K16/20), and H3.3 (H3F3B K23) being elevated following HIIT (*Figure 7A–D*). H1.5 K48 was exclusively quantified in post-HIIT samples, however, it was only quantified in four out of seven of the post-HIIT samples, which is our minimum cut-off for valid values. The regulatory roles of these HIIT-regulated histone acetyl-sites are unknown. However, acetylation of the N-terminal of H2B, including K16 and K20, has been linked to transcriptional activation (*Chen et al., 2014*; *Parra et al., 2006*), including the expression of NAD-metabolic genes in yeast (*Parra et al., 2006*). The role of H3 K23 is also obscure, however, its acetylation contributes to the development of cancer through the recruitment of the transcription factor transcription intermediary factor 1-alpha (TRIM24) and the nuclear receptor estrogen receptor alpha (ERS1) (*Lv et al., 2017*; *Tsai et al., 2010*), augmenting expression of phosphatidylinositol 4,5-biphosphate 3-kinase catalytic subunit alpha (PIK3CA), a subunit of phosphoionositide-3-kinase (PI3K), and phosphorylation of protein kinase B (AKT1) (*Lv et al., 2017*). Indeed, TRIM24 promotes the abundance of glycolytic and TCA cycle genes in transforming human mammary epithelial cells (*Pathiraja et al., 2015*), while ERS1 regulates mitochondrial biogenesis in skeletal muscle (*Ribas et al., 2016*). Whether H3 K23 regulates TRIM24, ERS1, PI3K, and AKT1 activity and influences metabolism in skeletal muscle remains unclear. Further studies are warranted

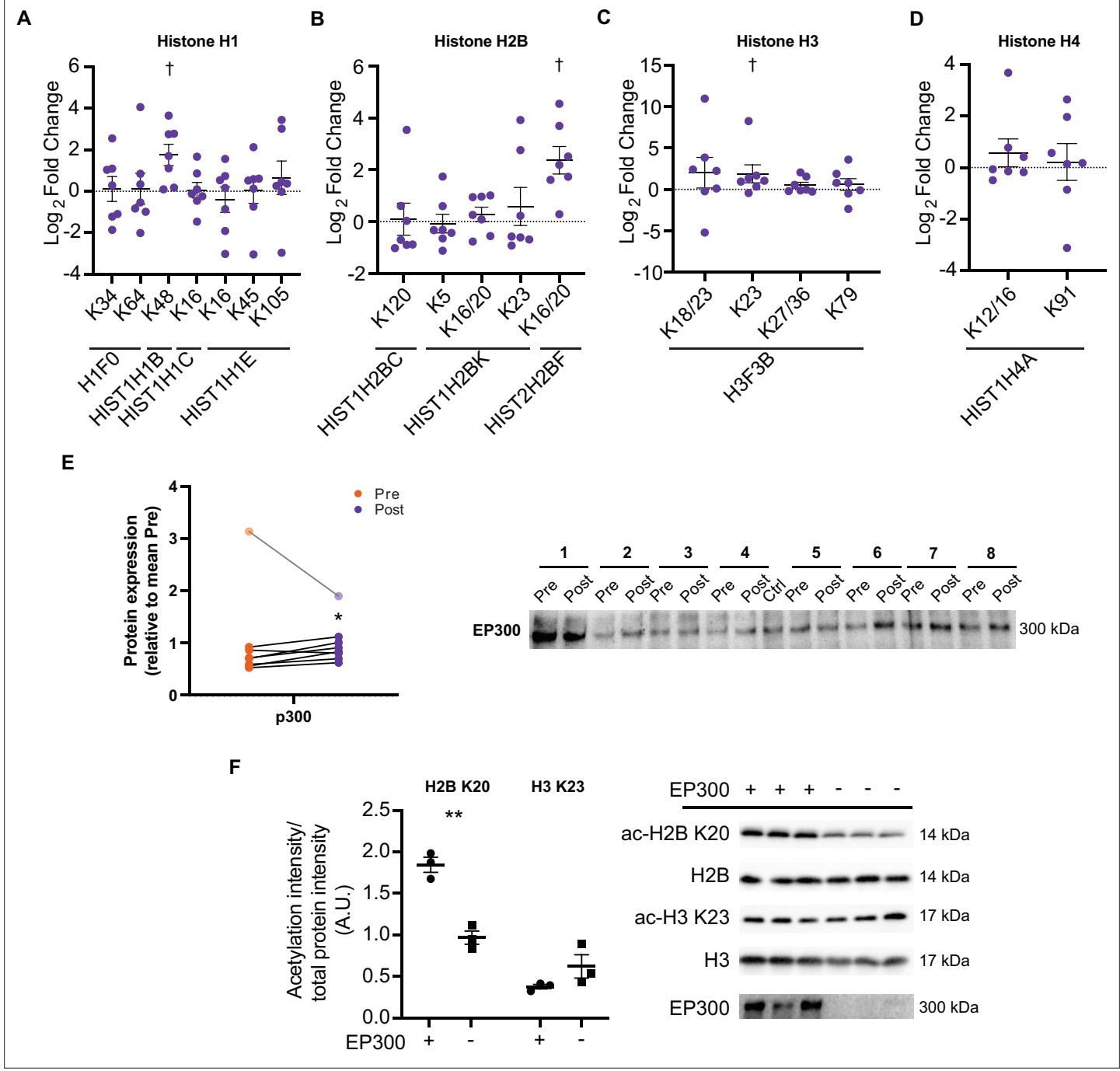

**Figure 7.** HIIT increases acetylation on specific histone acetyl-sites. HIIT increased acetylation on (**A**). H1.5 K48, (**B**). H2B type 2F K16/20 and (**C**). H3 K23. (**D**). HIIT did not alter H4 acetylation (n=7). (**E**) Immunoblotting analysis identified the upregulation of the nuclear-localized acetyltransferase EP300 (n=8, mean of three technical replicates; participant 1 was excluded as an outlier from statistical analysis (pre, post and fold-change values were all >3 median absolute deviations from the respective median), data for participant 1 is shown in the translucent data points). (**F**) Knockdown of EP300 reduces acetylation of H2B K20 but not H3 K23 in C2C12 myotubes (n=3). Representative images confirming equal loading are displayed in ***Figure 5—figure supplement 2***. Summary statistics are mean ± SEM. † ∏<0.05. * p<0.05, ** p<0.01, *** p<0.001.

The online version of this article includes the following source data and figure supplement(s) for figure 7:

**Source data 1.** Full image and annotation of H2B immunoblot.

**Source data 2.** Raw ImageLab file of H2B immunoblot.

**Source data 3.** Full image and annotation of H2B K20 immunoblot.

*Figure 7 continued on next page*

*Figure 7 continued*

**Source data 4.** Raw ImageLab file of H2B K20 immunoblot.

**Source data 5.** Raw quantification data of H2B K20/H2B immunoblots.

**Source data 6.** Full image and annotation of H3 immunoblot.

**Source data 7.** Raw ImageLab file of H3 immunoblot.

**Source data 8.** Full image and annotation of H3 K23 immunoblot.

**Source data 9.** Raw ImageLab file of H3 K23 immunoblot.

**Source data 10.** Raw quantification data of H3 K20/H3 immunoblots.

**Source data 11.** Full image and annotation of EP300 immunoblots (human skeletal muscle).

**Source data 12.** Raw ImageLab file of EP300 immunoblots (human skeletal muscle, replicate 1).

**Source data 13.** Raw ImageLab file of EP300 immunoblots (human skeletal muscle, replicates 2 & 3).

**Source data 14.** Raw quantification data of EP300 immunoblot (human skeletal muscle).

**Figure supplement 1.** Normalization of EP300.

**Figure supplement 1—source data 1.** Full image and annotation of EP300 (human skeletal muscle) replicate 1 stain-free blot.

**Figure supplement 1—source data 2.** Raw ImageLab file of EP300 (human skeletal muscle) replicate 1 stain-free blot.

**Figure supplement 1—source data 3.** Raw quantification data of EP300 (human skeletal muscle) replicate 1 with and without normalization.

to understand the transcriptional and metabolic implications of exercise training-induced acetylation on these histone sites.

Histone acetyltransferase p300 (EP300) is a nuclear-localized acetyltransferase targeting numerous histone acetylation sites (*Weinert et al., 2018*). Therefore, we hypothesized that EP300 may be involved in the regulation of histone acetylation following exercise training. Similarly to the upregulation of acetylation on H2B K16/20 and H3 K23, HIIT increased the protein abundance of EP300 by 23% (participant 1 was excluded as an outlier; *Figure 7E*). Given the small effect and the presence of a substantial outlier, we performed this particular analysis in triplicate. Additionally, to rule out that any small differences may be due to uneven loading, we normalized the first experimental replicate to a stain-free gel loading image (a loading image was only generated for the first replicate). EP300 displayed a similar increase whether normalized to the stain-free gel or not (29% vs 32% increase for normalized and non-normalized data, respectively; *Figure 7—figure supplement 1*). Next, to investigate the potential regulatory role of EP300 in the acetylation of these exercise-responsive histone sites in skeletal muscle, we knocked down EP300 in C2C12 myotubes and assessed the acetylation of H2B K20 and H3 K23 via immunoblotting. Knockdown of EP300 reduced acetylation of H2B K20 by 48%, whilst acetylation of H3 K23 was unaffected (*Figure 7F*). Together, these analyses identify EP300 as an exercise-training responsive acetyltransferase, which regulates the acetylation of H2B K20 in skeletal muscle.

## Conclusion

Here, we leveraged advances in DIA mass spectrometry to produce a deep human skeletal muscle proteome and applied this approach to ascertain exercise training adaptations. Furthermore, we provide the first investigation of the human acetylome in response to exercise training. In the proteome, we highlight known and novel exercise-responsive proteins, including a potential mechanism to explain altered skeletal muscle calcium sensitivity following HIIT. In the acetylome, we describe the predominant acetylation of mitochondrial proteins after HIIT, particularly of TCA cycle proteins and subunits of electron transport chain complex V, as well as highlighting the regulation of novel exercise-responsive histone acetyl-sites. Altogether, we provide a substantial hypothesis-generating resource, identifying novel exercise-regulated proteins and acetyl-sites with the aim of stimulating further mechanistic research investigating how exercise improves metabolic health.

## Materials and methods

### Participants

Participants were enrolled by AKL and MH between August and September 2017. Eight men were assessed for eligibility. All eight men were eligible and gave their informed consent after receiving written and oral information about the study and potential risks associated with the experimental procedures (*Figure 1—figure supplement 1*). Inclusion criteria were healthy males, 18–40 years old, $VO_{2max}$ between 45–55 mL·min$^{-1}$·kg$^{-1}$ or 3000–4000 mL·min$^{-1}$ and BMI between 19–26 kg·m$^{-2}$. Exclusion criteria were abnormal electrocardiogram, chronic disease, ongoing medical treatment, and smoking. Subject characteristics are presented in *Figure 1B*. The study was approved by the National Committee on Health Research Ethics (H-17004045) and registered at clinicaltrials.gov (NCT03270475). The study complied with the Declaration of Helsinki (2013).

### Study outcomes

The pre-determined primary outcome was the change from baseline in the skeletal muscle proteome. Secondary outcomes were changes from baseline in the skeletal muscle acetylome, skeletal muscle mitochondrial respiration and whole-body maximal oxygen consumption ($VO_{2max}$).

### Experimental trials

The study consisted of two experimental days separated by five weeks of high-intensity interval training (HIIT). At each experimental day, body composition was measured by dual-energy X-ray absorptiometry (DXA) scans (Lunar iDXA, GE Healthcare, GE Medical systems, Belgium). Then, subjects rested in a supine position for 15 min and two 4 mm incisions separated by 2 cm were made in the above the *vastus lateralis* muscle under local anesthesia (Xylocaine 20 mg·mL$^{-1}$ without epinephrine, Astra-Zeneca, London, UK) and a muscle biopsy was taken from each incision using a Bergström needle with suction. Afterwards, $VO_{2max}$ was determined by indirect calorimetry using a breath-by-breath gas analyser (Oxycon Pro, CareFusion, San Diego, USA) during an incremental test on a bike ergometer (Monark LC4, Monark Exercise, Vansbro, Sweden). The incremental test consisted of 5 min rest followed by 4 min stages at 50, 100, 150 W, after which the resistance was increased in increments of 30 W·min$^{-1}$ until exhaustion. Time to exhaustion was used to calculate incremental peak power output (iPPO) accounting for time spent at the last increment. Subjects completed the experimental protocol the same time of day before and after the intervention and were instructed to eat a small meal with 500 mL water two hours before arriving at the laboratory. In addition, subjects abstained from alcohol, caffeine, and physical activity for 48 hr before each experimental day and were instructed to maintain their activity level and diet during the study. The experimental days were performed three days before and after the first and last training, respectively. All experimental trials were performed at the Department of Nutrition, Exercise and Sports, University of Copenhagen. Experimental trials were completed in October 2017.

### Body composition

At each experimental day, two scans were performed to reduce scan-to-scan variation (*Nana et al., 2012*). Before the first scanning, subjects rested in a supine position for 10 min to allow fluid distribution reducing intra-scan variation (*Berg et al., 1993*; *Cerniglia et al., 2007*). Whole-body and leg composition were calculated using software (enCORE Forma v. 15, GE Healthcare Lunar, Buckinghamshire, UK).

### Substrate utilization

Gas exchange measurements during rest and submaximal exercise were used to calculate fat and glucose oxidative as previously described (*Jeukendrup and Wallis, 2005*) and corrected for protein oxidation (*Lee et al., 2013*). Fat oxidation and carbohydrate oxidation were calculated as:

$$Fat\ Oxidation\ \left(\tfrac{mg}{min}\right) = \left(1.695 \cdot \dot{V}O_2 - 1.701 \cdot \dot{V}CO_2\right) - \tfrac{1.92 \cdot 0.14 \cdot BM}{1440}$$

$$Carbohydrate\ Oxidation\ \left(\tfrac{mg}{min}\right) = \left(4.344 \cdot \dot{V}CO_2 - 3.061 \cdot \dot{V}O_2\right) - \tfrac{2.87 \cdot 0.14 \cdot BM}{1440}$$

where $VO_2$ and $VCO_2$ are in L·min$^{-1}$ and *BM (Body mass)* is in kg.

## Muscle biopsies

Immediately after collection, muscle biopsies were rinsed in ice-cold saline. A small piece (~15 mg) of the muscle was transferred into ice-cold preservation solution (BIOPS) for analysis of mitochondrial respiratory capacity. The remaining muscle tissue was snap-frozen in liquid nitrogen and stored at –80 °C. The frozen muscle samples were freeze-dried for 48 hr and dissected free from connective tissues, fat and blood under a microscope in a humidity-controlled room (~25% humidity) before storage at –80 °C for later analyses.

## High intensity interval training

Subjects underwent a five-week high-intensity interval training intervention consisting of three weekly training sessions with 4–5×4 min intervals interspersed by two min active recovery on indoor spinning bikes. The training volume increased from four intervals during the first two weeks to five intervals during the last three weeks. During each interval, subjects were instructed to reach >90% of maximal heart rate ($HR_{max}$). All training sessions were supervised and subjects received verbal encouragement during all intervals. The mean $HR_{max}$ during intervals was 96% ± 1% of $HR_{max}$.

## Mitochondrial profiling

Mitochondrial respiratory capacity was determined in duplicate by high-resolution respirometry (Oxygraph-2k, Oroboros Instruments, Innsbruck, Austria) at 37 °C using 1–3 mg wet weight muscle fibers per chamber in MiR06. Oxygen concentrations were kept between 200 and 450 µM throughout the experiment. The muscle sample in BIOPS was prepared for high-resolution respirometry as previously described (*Pesta and Gnaiger, 2012*). In short, muscle fibers were dissected free from connective tissue and fat followed by permeabilization in BIOPS with saponin and washed in mito-chondrial respiration buffer (MiR06). A substrate-uncoupler-inhibitor titration protocol was used to assess different respiratory states as previously described (*Pesta et al., 2011*). Leak respiration ($L_N$) in the absence of adenylates was induced by the addition of malate (2 mM) and octanoylcarnitine (0.2 mM). Fatty acid oxidation (FAO) was determined after titration of ADP (2.5 mM). Submaximal CI and CI +II-linked respiration ($CI_D$ and $P_D$) was measured in the presence of glutamate (10 mM) and succinate (10 mM), respectively. Oxidative phosphorylation capacity (P) was assessed after another titration of ADP (2.5 mM). The mitochondrial outer membrane integrity was then tested with the addition of cytochrome C (10 µM). The average change in oxygen flux was 3.8% (range: 0.2–9.4%). Respiratory electron transfer-pathway capacity (E) was measured during step-wise (0.5 µM) titration of carbonyl cyanide p-trifluoro-methoxyphenyl hydrazine (FCCP). Succinate-supported electron transfer-pathway capacity ($E_{CII}$) was assessed by the addition of rotenone (0.5 µM). Residual oxygen consumption was determined after the addition of malonic acid (5 mM), myxothaizol (0.5 µM), and antimycin A (2.5 µM) and subtracted from each respiratory state.

Maximal enzyme activity of citrate synthase was determined from muscle homogenate using fluorometry (Fluoroscan Ascent, Thermo Fisher Scientific, Waltham, USA) at 25 °C as previously described (*Lowry, 2012*).

## Proteomic sample preparation

Human skeletal muscles were lysed in a buffer consisting of 1% Sodium Deoxycholate (SDC), 10 mM Tris (2-carboxyethyl) phosphine (TCEP), 40 mM Chloroacetamide (CAA) and 100 mM of Tris (pH 8.5). The muscles were homogenized with an Ultra Turrax homogenizer (IKA). Lysates were boiled at 95 °C for 10 min and sonicated using a tip sonicator. Lysates were then centrifuged at 16,000 g for 10 min. Proteins were digested using the endoproteinases LysC and trypsin (1:100 w/w) at 37 °C overnight with shaking. Digested peptides were acidified using 1% Trifluoroacetic acid (TFA) and precipitated SDC was removed by centrifugation. Peptides were purified using Sep-pak C18 cartridges (Waters) and were eluted in 50% acetonitrile. Twenty µg of peptides were saved for total proteome analysis while 3 mg peptides were saved for enrichment of acetylated peptides.

## Acetylated peptide enrichment

The acetyl peptide enrichment was performed on 3 mg of digested peptides using PTMscan Acetyl-Lysine Motif kit (Cell Signaling #13416). The peptides were mixed with 100 µL of 10 x IP buffer (500 mM MOPS, pH 7.2, 100 mM Na-phosphate, 500 mM NaCl, 5% NP-40). The acetonitrile was

removed and the volume was reduced to ~1 mL by vacuum centrifugation. The final volume was adjusted with water to a concentration of 3 mg·mL$^{-1}$. The peptides were clarified by centrifugation at 20,000 g for 5 min. Twenty µL of anti-acetylated lysine antibody was washed 3 times in 1 mL IP buffer and then mixed with the peptides. Peptides were enriched overnight at 4 °C, washed 3 times in 1 mL ice-cold (4 °C) IP buffer, 4 times in 1 mL ice-cold IP buffer without NP-40, and once in 1 mL water. All wash buffer was removed via aspiration. Acetylated peptides were eluted with 100 µL 0.15% TFA. The elution procedure was repeated three times. The eluted peptides were de-salted on C18 stage-tips (*Rappsilber et al., 2007*), vacuum concentrated and resuspended in buffer A* (5% acetonitrile, 0.1% trifluoroacetic acid).

## High pH reversed-phase fractionation

To generate a deep proteome library for data-independent acquisition (DIA), 50 µg of peptides were pooled from pre- and post-HIIT samples, separately, and fractionated using high pH reversed-phase chromatography (*Kulak et al., 2017*). Twenty-four fractions (per pool) were automatically concatenated using a rotor valve shift of 90 s. Approximately 0.3 µg of each fraction were subjected to LC-MS/MS measurements via data-dependent acquisition (DDA). To obtain comprehensive coverage of the acetylome, acetylated peptides from each sample were fractionated into 8 fractions using a rotor valve shift of 90 s. For the first subject (both pre and post samples), all eight fractions were analyzed separately. After observing lower than expected peptide intensities for the first subject, the neighboring two factions were pooled together for each of the remaining samples (total four fractions/subject/timepoint) before LC-MS analysis.

## Mass spectrometry

Peptides were measured using LC-MS instrumentation consisting of an Easy nanoflow HPLC system (Thermo Fisher Scientific, Bremen, Germany) coupled via a nanoelectrospray ion source (Thermo Fischer Scientific, Bremen, Germany) to a Q Exactive HF-X mass spectrometer. Purified peptides were separated on a 50 cm C18 column (inner diameter 75 µm, 1.8 µm beads, Dr. Maisch GmbH, Germany). Peptides were loaded onto the column with buffer A (0.5% formic acid) and eluted with a 100 min linear gradient increasing from 2% to 40% buffer B (80% acetonitrile, 0.5% formic acid). After the gradient, the column was washed with 90% buffer B and re-equilibrated with buffer A. Peptides were measured in singleton.

For the deep proteome library generation, mass spectra were acquired from each fraction via DDA with automatic switching between MS and MS/MS using a top 15 method. MS spectra were acquired in the Orbitrap analyzer with a mass range of 300–1750 m/z and 60,000 resolutions at m/z 200 with a target of $3 \times 10^6$ ions and a maximum injection time of 25ms. HCD peptide fragments acquired at 27 normalized collision energy were analyzed at 15,000 resolution in the Orbitrap analyzer with a target of $1 \times 105$ ions and a maximum injection time of 28 ms. The mass spectra for the acetylome experiments were also acquired from each fraction via DDA with a similar MS method used for deep proteome library generation, except the maximum injection time for the fragment ion spectra was set to 250ms. A DIA MS method was used for the total proteome measurements in which one full scan (300–1650 *m/z*, resolution = 60,000 at 200 *m/z*) at a target of $3 \times 10^6$ ions was first performed, followed by 32 windows with a resolution of 30,000 where precursor ions were fragmented with higher-energy collisional dissociation (stepped collision energy 25%, 27.5%, 30%) and analyzed with an AGC target of $3 \times 10^6$ ions and maximum injection time at 54ms in profile mode using positive polarity.

## Data processing

Raw MS files from the experiments measured in DDA mode (muscle proteome library & acetylome), were processed using MaxQuant (*Cox and Mann, 2008*). MS/MS spectra were searched by the Andromeda search engine (integrated into MaxQuant) against the decoy UniProt-human database (downloaded in December 2017) with forward and reverse sequences. In the main Andromeda search, precursor and fragment mass were matched with an initial mass tolerance of 6 ppm and 20 ppm, respectively. The search included variable modifications of methionine oxidation and N-terminal acetylation and fixed modification of carbamidomethyl cysteine. For the raw files from the acetylome experiments, acetylated lysine was also added as a variable modification. Acetylated peptides were filtered for a minimum Andromeda score of 40, as per the default settings for modified peptides. The

false discovery rate (FDR) was estimated for peptides and proteins individually using a target-decoy approach allowing a maximum of 1% false identifications from a revered sequence database. Raw files acquired in the DIA mode (total proteome) were processed using Biognosys Spectronaut software version 13 (*Bruderer et al., 2015*). A single peptide library was generated in Spectronaut using the combined MaxQuant search results for the DDA runs from both of the fractionated muscle samples. The experimental DIA runs were then analyzed in Spectronaut using default settings.

## Bioinformatics

Bioinformatic analyses were performed in the Perseus software (*Tyanova et al., 2016*). Categorical annotations were supplied in the form of Gene Ontology (GO) biological process (BP), molecular function (MF), and cellular component (CC), as well as UniProt Keywords, KEGG and REACTOME pathways. All annotations were extracted from the UniProt database. Quantified proteins were filtered to have at least 4 valid values in at least one time point. Missing data were imputed by drawing random numbers from a Gaussian distribution with a standard deviation of 30% and a downshift of 1.8 standard deviations from the mean. The imputed values have been tuned in order to simulate the distribution of lowly abundant proteins. To identify differentially regulated proteins and acetyl-sites, two-tailed paired *t*-tests were performed with a permutation-based false discovery correction applied (FDR = 0.05). In the acetylome, an additional analysis was performed to expand the identification of differentially expressed proteins, whereby an a posteriori information fusion scheme combining the biological relevance (fold change) and the statistical significance (p-value), was implemented, as previously described (*Xiao et al., 2014*). A ∏-value significance score cut-off of 0.05 was selected. Alterations to organelles were assessed using summed total protein abundances (*Wiśniewski, 2017*) and two-tailed paired t-tests (p<0.05). Comparisons between individual proteins measured via immunoblotting were performed using two-tailed paired t-tests (p<0.05). Hierarchical clustering analysis was performed on z-scored differentially regulated proteins using Euclidean distance. Enrichment analyses were performed on differentially regulated proteins and acetyl-sites using Fisher's exact tests and the application of a Benjamini-Hochberg false discovery correction (FDR = 0.02), as is the default setting in the Perseus software (*Tyanova et al., 2016*). Motif enrichment was performed on upregulated acetyl-sites using iceLogo (p<0.01) (*Colaert et al., 2009*). One-dimensional enrichment analyses were performed as previously described (*Cox and Mann, 2012*). To analyze the relative stoichiometries of acetylated proteins in *Figure 4* abundance corrected intensities (ACIs) were calculated for each acetyl-peptide by dividing the acetyl-peptide intensity by the intensity of the corresponding protein in the proteome. Enrichment analyses were performed against a background of all quantified proteins or acetyl-sites, except for hierarchical clustering analyses where clusters were compared within the z-scored differentially regulated matrix. To visualize complexes and functional protein-interactions in *Figure 6*, protein interaction networks were mapped using the STRING database, using an overall confidence cutoff of 0.7 without limiting the number of interactions (*Szklarczyk et al., 2019*), and further processed with Cytoscape (https://www.cytoscape.org).

## Cell culture

The mouse skeletal muscle cell line C2C12 (CRL1772, ATCC) was seeded into six-well plates at a density of $10^5$ cells per well. Cells were cultured in growth medium (Dulbeccoo's modified Eagle's medium containing 25 mM glucose, supplemented with 10% fetal bovine serum (FBS) and 1%. Penicillin-streptomycin) at 37 °C and 5% $CO_2$. Once the cells reached ~90% confluency, differentiation was induced by replacing the FBS with 2% horse serum. On day 4 of differentiation, cells were transfected with 25 mM EP300 siRNA or scramble control siRNA via the TransIT-X2 Dynamic Delivery System (Mirus Bio), following the manufacturer's instructions. Cells were harvested in 2% SDS on Day 6. Samples were immediately boiled at 95 °C, sonicated in a Diagenode Bioruptor water bath sonicator (15 × 30 s intervals), and centrifuged at 16,000 g for 10 min. The pellet was discarded and the supernatant was taken and used for immunoblotting analyses.

## Immunoblotting

For human skeletal muscle immunoblot analyses, a fresh batch of ice-cold homogenization buffer (10% glycerol, 20 mM Na-pyrophosphate, 150 mM NaCl, 50 mM HEPES (pH 7.5), 1% NP-40, 20 mM β-glycerophosphate, 2 mM $Na_3VO_4$, 10 mM NaF, 2 mM PMSF, 1 mM EDTA (pH 8), 1 mM EGTA (pH 8),

10 µg·mL⁻¹ Aprotinin, 10 µg·mL⁻¹ Leupeptin and 3 mM Benzamidine) was prepared and 80 µL added to 1 mg of freeze-dried and dissected muscle tissue. The tissue was homogenized for 1 min at 28.5 Hz in a TissueLyser (Qiagen TissueLyser II, Retsch GmbH, Germany) before samples were rotated end over end for 1 hr at 4 °C and centrifuged at 18,320 g for 20 min at 4 °C. The pellet was discarded and the supernatant was taken and used for immunoblotting analyses. Total protein concentration in the samples was determined with a standard BSA kit (Millipore). A total of 15 µL of the sample was diluted with 60 µL ultrapure water in order to keep the concentration within the linear range of the calibration curve (0.2–2 µg·µL⁻¹). The protein concentration of each sample was adjusted by the addition of 6×Laemmli buffer (7 mL 0.5 M Tris-base, 3 mL glycerol, 0.93 g DTT, 1 g SDS and 1.2 mg bromophenol blue) and ultrapure water to reach equal concentrations (1.5 µg total protein·µL⁻¹). Human skeletal muscle standard samples (Ctrl) were obtained as a pool of all samples included in the experiment.

Equal amounts of total protein were loaded on 10% TGX Stain-Free, 4–20% TGX Stain-Free, or 16.5% Criterion Tris-Tricine/Peptide gels (Bio-Rad) alongside two protein markers (Precision plus all blue and dual color standards, Bio-Rad), three human skeletal muscle control samples and a four-point standard curve (skeletal muscle samples of increasing volume). After standard SDS-page gel electrophoresis proteins were semi-dry transferred to a PVDF membrane. Membranes were blocked with either 2% skimmed milk or 3% BSA in Tris-buffered saline with 0.1% Tween-20 (TBST) before overnight incubation with primary antibody. Afterwards, the membranes were washed in TBST, incubated for 1 hr in HRP conjugated secondary antibody at room temperature and washed 3×15 min in TBST before the bands were visualized with an enhanced chemiluminescent reaction (Immobilon Forte Western HRP substrate, Millipore) and signals recorded with a digital camera (ChemiDoc MP Imaging System, Bio-Rad). Densitometry quantification of the immunoblotting band intensities was done using Image Lab version 4 (Bio-Rad) and determined as the total band intensity adjusted for the background intensity. The standard curve on each gel was used to confirm that the loaded amount of protein in samples was capable of determining differences between samples by the signal intensity being on the linear part of the standard curve. The variation of the triplicate human standard sample signal loaded was used to evaluate equal transfer of proteins across each gel. To confirm equal loading from the different samples, Coomasie staining and stain-free blot images were taken for representative immunoblots of each batch of prepared lysate (*Figure 5—figure supplement 2*). Immunoblotting of biological samples was performed in singleton, with the exception of EP300 which was performed in technical triplicates to confirm the presence of a biological outlier (participant 1).

Listed are the primary antibodies used and, where appropriate, their size of migration. Acetyl-lysine: #9441, Cell Signaling Technologies; EP300: #sc32244 (human skeletal muscle); #sc48343 (C2C12s), Santa Cruz Biotechnology, 300 kDa; ac-H2B K20: #ab177430, Abcam, 14 kDa; H2B: #12364, Cell Signaling Technologies, 14 kDa; ac-H3 K23: #14932, Cell Signaling Technologies, 17 kDa; H3: #4499, Cell Signaling Technologies, 17 kDa; MYLK2: PA5-29324, Invitrogen, 65 kDa; OXPHOS antibody cocktail: ab110411, Abcam, 15–55 kDa; SIRT1: 07–131, Millipore, 80 kDa; SIRT3: #5490, Cell Signaling Technologies, 28 kDa; SIRT5: #8782, Cell Signaling Technologies, 30 kDa; SIRT6: #12486, Cell Signaling Technologies, 36–42 kDa; ac-SOD2 K68: #ab13737, Abcam, 24 kDa; ac-SOD2 K122: #ab214675, Abcam, 24 kDa. The anti-GCN5L1 antibody (15 kDa) was a kind gift from Iain Scott, PhD, University of Pittsburgh (*Scott et al., 2012*; *Thapa et al., 2020*). HRP conjugated goat anti-rabbit (4010–05, SouthernBiotech) and a goat anti-mouse (P0447, DAKO Denmark) were used as secondary antibodies.

## Statistical analyses

For non-omics analyses, two-tailed paired t-tests were applied to assess the effect of HIIT, except for substrate oxidation analyses whereby a mixed linear model was performed. Independent samples two-tailed t-tests were used to assess the effect of EP300 KD on histone acetylation. Significance was accepted as $p < 0.05$.

## Acknowledgements

This work was supported by an unconditional donation from the Novo Nordisk Foundation (NNF) to NNF Center for Basic Metabolic Research (http://www.cbmr.ku.dk) (Grant number NNF18CC0034900), NNF Center for Protein Research (https://www.cpr.ku.dk/) (Grant number NNF14CC001), and Team Denmark. We acknowledge Matthias Mann, Jens Jung Nielsen, Rebeca Soria Romero and the mass

spectrometry platform from at the NNF Center for Protein Research for technical assistance and access to mass spectrometers. We thank Iain Scott from the University of Pittsburgh Department of Medicine for the kind gift of the anti-GCN5L1 antibody.

## Additional information

### Funding

| Funder | Grant reference number | Author |
|---|---|---|
| NNF Center for Basic Metabolic Research | NNF18CC0034900 | Atul Shahaji Deshmukh |
| NNF Center for Protein Research | NNF14CC001 | Atul Shahaji Deshmukh |
| Team Denmark | | Morten Hostrup |

The funders had no role in study design, data collection and interpretation, or the decision to submit the work for publication.

### Author contributions

Morten Hostrup, Atul Shahaji Deshmukh, Conceptualization, Data curation, Formal analysis, Funding acquisition, Investigation, Methodology, Project administration, Resources, Supervision, Validation, Writing – original draft, Writing – review and editing; Anders Krogh Lemminger, Conceptualization, Data curation, Formal analysis, Investigation, Methodology, Validation, Writing – original draft, Writing – review and editing; Ben Stocks, Data curation, Formal analysis, Investigation, Methodology, Validation, Visualization, Writing – original draft, Writing – review and editing; Alba Gonzalez-Franquesa, Julia Prats Quesada, Investigation, Methodology, Writing – review and editing; Jeppe Kjærgaard Larsen, Formal analysis, Investigation, Methodology, Writing – review and editing; Martin Thomassen, Formal analysis, Investigation, Validation, Writing – review and editing; Brian Tate Weinert, Writing – review and editing; Jens Bangsbo, Conceptualization, Funding acquisition, Methodology, Resources, Writing – review and editing

### Author ORCIDs

Morten Hostrup http://orcid.org/0000-0002-6201-2483
Ben Stocks http://orcid.org/0000-0002-9281-0793
Martin Thomassen http://orcid.org/0000-0001-7503-6815
Atul Shahaji Deshmukh http://orcid.org/0000-0002-2278-1843

### Ethics

Clinical trial registration NCT03270475.
Eight men gave their informed consent after receiving written and oral information about the study and potential risks associated with the experimental procedures. The study was approved by the National Committee on Health Research Ethics (H-17004045) and registered at clinicaltrials.gov (NCT03270475). The study complied with the Declaration of Helsinki (2013).

### Decision letter and Author response

Decision letter https://doi.org/10.7554/eLife.69802.sa1
Author response https://doi.org/10.7554/eLife.69802.sa2

## Additional files

### Supplementary files

- Supplementary file 1. Identified proteins.
- Supplementary file 2. Quantified proteins and their regulation by HIIT.
- Supplementary file 3. Fisher's exact test of HIIT-regulated proteins (FDR <0.05).
- Supplementary file 4. Identified acetyl-sites.

- Supplementary file 5. Quantified acetyl-sites and their regulation by HIIT.
- Supplementary file 6. Summed acetylation intensity per protein.
- Supplementary file 7. One-dimensional enrichment analysis of summed acetylation intensity.
- Supplementary file 8. Abundance corrected intensities (ACIs) of acetyl-sites pre-HIIT.
- Supplementary file 9. One-dimensional enrichment analysis of pre-HIIT acetyl-site ACI (Leading protein ID was used for relative enrichment).
- Supplementary file 10. Fisher's exact test of HIIT-regulated (∏<0.05) acetyl-sites (Leading protein ID was used for relative enrichment).
- Transparent reporting form

### Data availability

Source data for the proteome and acetylome in Figures 2-7 and S1-S4 has been uploaded to PRIDE with the dataset identifier PXD023084.

The following dataset was generated:

| Author(s) | Year | Dataset title | Dataset URL | Database and Identifier |
|-----------|------|---------------|-------------|-------------------------|
| Deshmukh AS | 2022 | High-intensity interval training remodels the proteome and acetylome of human skeletal muscle | https://www.ebi.ac.uk/pride/archive/projects/PXD023084 | PRIDE, PXD023084 |

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

# Appendix 1

### Appendix 1—key resources table

| Reagent type (species) or resource | Designation | Source or reference | Identifiers | Additional information |
|---|---|---|---|---|
| Antibody | MYLK2 Antibody, rabbit polyclonal | Invitrogen | Cat#PA5-29324 RRID: AB_2546800 | (1:1000) |
| Antibody | Total OXPHOS Human WB Antibody Cocktail, mouse monoclonal | Abcam | Cat#ab110411 RRID: AB_2756818 | (1:1000) |
| Antibody | Sirt1(Sir2) Antibody, rabbit polyclonal | Millipore Sigma | Cat#07–131 RRID: AB_10067921 | (1:1000) |
| Antibody | Sirt3 (D22A3) Antibody, rabbit monoclonal | Cell Signaling Technologies | Cat#5490 RRID: AB_10828246 | (1:2000) |
| Antibody | Sirt5 Antibody, rabbit monoclonal | Cell Signaling Technologies | Cat#8782 RRID: AB_2716763 | (1:1000) |
| Antibody | Sirt6 Antibody, rabbit monoclonal | Cell Signaling Technologies | Cat#12486 RRID: AB_2636969 | (1:1000) |
| Antibody | Acetyl-lysine Antibody, rabbit polyclonal | Cell Signaling Technologies | Cat#9441 RRID: AB_331805 | (1:1000) |
| Antibody | Ac-SOD2 K68 Antibody, rabbit monoclonal | Abcam | Cat#ab13737 RRID: AB_2784527 | (1:1000) |
| Antibody | Ac-SOD2 K122 Antibody, rabbit monoclonal | Abcam | Cat#ab214675 RRID: AB_2892634 | (1:1000) |
| Antibody | P300 Mouse Antibody, mouse monoclonal | Santa Cruz Biotechnologies | Cat#sc48343 RRID: AB_628075 | (1:200) |
| Antibody | P300 Mouse Antibody, mouse monoclonal | Santa Cruz Biotechnologies | Cat#sc32244 RRID: AB_628076 | (1:500) |
| Antibody | Ac-H2B K20 Antibody, rabbit monoclonal | Abcam | Cat#ab177430 | (1:1000) |
| Antibody | H2B Antibody, rabbit monoclonal | Cell Signaling Technologies | Cat#12364 RRID: AB_2714167 | (1:3000) |
| Antibody | Ac-H3 K23 Antibody, rabbit monoclonal | Cell Signaling Technologies | Cat#14932 RRID: AB_2798650 | (1:1000) |
| Antibody | H3 Antibody, rabbit monoclonal | Cell Signaling Technologies | Cat#4499 RRID: AB_10544537 | (1:3000) |
| Antibody | Goat Anti-Rabbit Ig, Human ads-HRP, goat polyclonal | Southern Biotech | Cat#4010–05 RRID: AB_2632593 | (1:5000) |
| Antibody | Goat Anti-Mouse, goat polyclonal | Dako (Agilent) | Cat# P0447 RRID: AB_2617137 | (1:5000) |
| Biological sample (human) | Human skeletal muscle (*vastus lateralis*) | This paper | | Not externally available |
| Cell line (mouse) | C2C12 (ATCC CRL-1772) | ATCC | Cat#1722 RRID: CVCL_0188 | Authenticated by ATCC – CO1 assay. Tested in-house for mycoplasma – negative. |
| Chemical compound, drug | Glycerol | Millipore Sigma | Cat# G5516; CAS# 56-81-5 | |
| Chemical compound, drug | Sodium pyrophosphate decahydrate | Millipore Sigma | Cat# 221368; CAS# 13472-36-1 | |
| Chemical compound, drug | Sodium chloride | Millipore Sigma | Cat# S9888; CAS# 7647-14-5 | |

*Appendix 1 Continued on next page*

*Appendix 1 Continued*

| Reagent type (species) or resource | Designation | Source or reference | Identifiers | Additional information |
|---|---|---|---|---|
| Chemical compound, drug | HEPES | Millipore Sigma | Cat# H3375; CAS# 7365-45-9 | |
| Chemical compound, drug | IGEPAL CA-630, NP-40 | Millipore Sigma | Cat# I8896; CAS# 9002-93-1 | |
| Chemical compound, drug | β-Glycerophosphate disodium salt hydrate | Millipore Sigma | Cat# G9422; CAS# 154804-51-0 | |
| Chemical compound, drug | Sodium orthovanadate | Millipore Sigma | Cat# S6508; CAS# 13721-39-6 | |
| Chemical compound, drug | Sodium fluoride | Millipore Sigma | Cat# S1504; CAS# 7681-49-4 | |
| Chemical compound, drug | Phenylmethanesulfonyl fluoride | Millipore Sigma | Cat# P7626; CAS# 329-98-6 | |
| Chemical compound, drug | EDTA (for electrophoresis) | Millipore Sigma | Cat# E5134; CAS# 6381-92-6 | |
| Chemical compound, drug | EGTA | Millipore Sigma | Cat# E4378; CAS# 67-42-5 | |
| Chemical compound, drug | Aprotinin from bovine lung | Millipore Sigma | Cat# A1153; CAS# 9087-70-1 | |
| Chemical compound, drug | Leupeptin | Millipore Sigma | Cat# L2884; CAS# 103476-89-7 | |
| Chemical compound, drug | Benzamidine hydrochloride hydrate | Millipore Sigma | Cat# B6506; CAS# 206752-36-5 | |
| Chemical compound, drug | Bovine serum albumin | Millipore Sigma | Cat# A7906; CAS# 9048-46-8 | |
| Chemical compound, drug | Trizma base | Millipore Sigma | Cat# T1503; CAS# 77-86-1 | |
| Chemical compound, drug | DL-Dithiothreitol | Millipore Sigma | Cat# D0632 and 43819; CAS# 3483-12-3 | |
| Chemical compound, drug | Sodium dodecyl sulfate | Millipore Sigma | Cat# L3771; CAS# 151-21-3 | |
| Chemical compound, drug | Bromophenol Blue sodium salt | Millipore Sigma | Cat# B8026; CAS# 34725-61-6 | |
| Chemical compound, drug | Tween 20 | Millipore Sigma | Cat# P9416; CAS# 9005-64-5 | |
| Chemical compound, drug | Skimmed milk powder | Easis | Cat# 801,300 | |

*Appendix 1 Continued on next page*

*Appendix 1 Continued*

| Reagent type (species) or resource | Designation | Source or reference | Identifiers | Additional information |
|---|---|---|---|---|
| Chemical compound, drug | Glycine | Millipore Sigma | Cat# G7126; CAS# 56-40-6 | |
| Chemical compound, drug | 2-propanol | Millipore Sigma | Cat# 109634; CAS# 67-63-0 | |
| Chemical compound, drug | Ethanol 96% | Plum | Cat# 201146; CAS# 64-17-5 | |
| Chemical compound, drug | Brilliant Blue R | Millipore Sigma | Cat# B7920; CAS# 6104-59-2 | |
| Chemical compound, drug | Acetic acid (glacial) 100% | Millipore Sigma | Cat# 100063; CAS# 64-19-7 | |
| Chemical compound, drug | Precision plus all blue | BioRad | Cat# 1610373 | |
| Chemical compound, drug | Precision plus dual color | BioRad | Cat# 1610374 | |
| Chemical compound, drug | Immobilon Forte Western HRP substrate | Millipore Sigma | Cat# WBLUF 0500 | |
| Chemical compound, drug | Bovine Serum Albumin Standard | Thermo Fisher Scientific | Cat# 23,209 | |
| Chemical compound, drug | BCA Protein Assay Reagent A | Thermo Fisher Scientific | Cat# 23,223 | |
| Chemical compound, drug | BCA Protein Assay Reagent B | Thermo Fisher Scientific | Cat# 23,224 | |
| Chemical compound, drug | Potassium phosphate dibasic trihydrate | Millipore Sigma | Cat# P9666; CAS# 16788-57-1 | |
| Chemical compound, drug | Potassium dihydrogen phosphate | Millipore Sigma | Cat# 1.04873; CAS# 7778-77-0 | |
| Chemical compound, drug | BCA Protein Assay Kit | Thermo Fisher Scientific | Cat# 23,225 | |
| Chemical compound, drug | Trizma hydrochloride | Millipore Sigma | Cat# T3253; CAS# 1185-53-1 | |
| Chemical compound, drug | EDTA | Millipore Sigma | Cat# E9884; CAS# 60-00-4 | |
| Chemical compound, drug | NAD | Millipore Sigma | Cat# NAD100-RO; CAS# 53-84-9 | |
| Chemical compound, drug | L-(−)-Malic acid sodium salt | Millipore Sigma | Cat# M1125; CAS# 68303-40-2 | |

*Appendix 1 Continued on next page*

| Reagent type (species) or resource | Designation | Source or reference | Identifiers | Additional information |
|---|---|---|---|---|
| Chemical compound, drug | Acetyl-Coenzyme A | Millipore Sigma | Cat# ACOA-RO | |
| Chemical compound, drug | L-Malate Dehydrogenase (L-MDH) | Millipore Sigma | Cat# LMDH-RO | |
| Chemical compound, drug | NADH | Millipore Sigma | Cat# 10107735001 | |
| Chemical compound, drug | Sucrose | Millipore Sigma | Cat# S7903; CAS# 57-50-1 | |
| Chemical compound, drug | Magnesium chloride hexahydrate | Millipore Sigma | Cat# M2670; CAS# 7791-18-6 | |
| Chemical compound, drug | Taurine | Millipore Sigma | Cat# T0625; CAS# 107-35-7 | |
| Chemical compound, drug | Potassium Dihydrogen Phosphate | Millipore Sigma | Cat# P9791; CAS# 7778-77-0 | |
| Chemical compound, drug | HEPES | Millipore Sigma | Cat# H3375; CAS# 7365-45-9 | |
| Chemical compound, drug | Bovine serum albumin (fatty acid free) | Millipore Sigma | Cat# A7030; CAS# 9048-46-8 | |
| Chemical compound, drug | Catalase from bovine liver | Millipore Sigma | Cat# C9322; CAS# 9001-05-2 | |
| Chemical compound, drug | Lactobionic acid | Millipore Sigma | Cat# 153516; CAS# 96-82-2 | |
| Chemical compound, drug | MES potassium salt | Millipore Sigma | Cat# M0895; CAS# 39946-25-3 | |
| Chemical compound, drug | ATP | Millipore Sigma | Cat# A3377; CAS# 34369-07-8 | |
| Chemical compound, drug | Sodium creatine phosphate dibasic tetrahydrate | Millipore Sigma | Cat# 27920; CAS# 71519-72-7 | |
| Chemical compound, drug | Imidazole | Millipore Sigma | Cat# I0250; CAS# 288-32-4 | |
| Chemical compound, drug | Potassium hydroxide | Millipore Sigma | Cat# P1767; CAS# 1310-58-3 | |
| Chemical compound, drug | Calcium carbonate | Millipore Sigma | Cat# C4830; CAS# 471-34-1 | |
| Chemical compound, drug | Saponin from quillaja bark | Millipore Sigma | Cat# S7900; CAS# 8047-15-2 | |
| Chemical compound, drug | L-(−)-Malic acid sodium salt | Millipore Sigma | Cat# M1125; CAS# 68303-40-2 | |

| Reagent type (species) or resource | Designation | Source or reference | Identifiers | Additional information |
|---|---|---|---|---|
| Chemical compound, drug | Octanoyl-L-carnitine | Millipore Sigma | Cat# 50892; CAS# 25243-95-2 | |
| Chemical compound, drug | ADP | Millipore Sigma | Cat# A5285; CAS# 72696-48-1 | |
| Chemical compound, drug | L-glutamic acid monosodium salt monohydrate | Millipore Sigma | Cat# 49621; CAS# 6106-04-3 | |
| Chemical compound, drug | Sodium succinate dibasic hexahydrate | Millipore Sigma | Cat# S2378; CAS# 6106-21-4 | |
| Chemical compound, drug | Cytochrome c from equine heart | Millipore Sigma | Cat# C2506; CAS# 9007-43-6 | |
| Chemical compound, drug | FCCP | Millipore Sigma | Cat# C2920; CAS# 370-86-5 | |
| Chemical compound, drug | Rotenone | Millipore Sigma | Cat# R8875; CAS# 83-79-4 | |
| Chemical compound, drug | Malonic acid | Millipore Sigma | Cat# M1296; CAS# 141-82-2 | |
| Chemical compound, drug | Myxothiazol | Millipore Sigma | Cat# T5580; CAS# 76706-55-3 | |
| Chemical compound, drug | Antimycin A from Streptomyces sp. | Millipore Sigma | Cat# A8674; CAS# 1397-94-0 | |
| Chemical compound, drug | Sodium deoxycholate | Sigma-Aldrich | Cat# D6750 CAS# 302-95-4 | |
| Chemical compound, drug | Tris(2-carboxyethyl) phosphine hydrochloride | Sigma-Aldrich | Cat# C4706 CAS# 51805-45-9 | |
| Chemical compound, drug | 2-Chloroacetamide | Sigma-Aldrich | Cat# 22,790 CAS# 79-07-2 | |
| Chemical compound, drug | Trypsin | Sigma-Aldrich | Cat# T7575 | |
| Chemical compound, drug | Lysine | Wako | Cat# 124–06871 | |
| Chemical compound, drug | DMEM high glucose | Gibco | Cat# 41965039 | |
| Chemical compound, drug | Fetal bovine serum(FBS) | Sigma-Aldrich | Cat# F7524 | |
| Chemical compound, drug | Penicillin-Streptomycin, liquid | Gibco | Cat# 15070–063 | |
| Chemical compound, drug | Opti-MEM I Reduced Serum Medium | Gibco | Cat# 31985062 | |

| Reagent type (species) or resource | Designation | Source or reference | Identifiers | Additional information |
|---|---|---|---|---|
| Chemical compound, drug | TransIT-X2 Dynamic Delivery System | Mirus Bio | Cat# MIR6003 | |
| Sequence-based reagent | EP300 siRNA | Merck Sigma | Cat# SASI_Mm01_00159721 | |
| Commercial assay or kit | PTMScan Acetyl-Lysine Motif [Ac-K] Kit | Cell Signaling Technologies | Cat#13,416 | |
| Software, algorithm, | Graphpad Prism 8.0 | https://www.graphpad.com/ | RRID: SCR_002798 | |
| Software, algorithm | MaxQuant 1.5.3.30 | https://maxquant.org/ | RRID: SCR_014485 | |
| Software, algorithm | Spectronaut v14 | https://biognosys.com/shop/spectronaut | | |
| Software, algorithm | Perseus 1.6.10.50 | http://www.coxdocs.org/doku.php?id=perseus:start | RRID: SCR_015753 | |
| Software, algorithm | R Studio | https://rstudio.com/ | RRID: SCR_000432 | |
| Software, algorithm | Cytoscape v3.7.2 | https://www.cytoscape.org | RRID: SCR_003032 | |
| Software, algorithm | iceLogo | https://iomics.ugent.be/icelogoserver/ | RRID: SCR_012137 | |
| Software, algorithm | Adobe Illustrator 24.3 | https://www.adobe.com/products/illustrator | RRID: SCR_010279 | |
| Software, algorithm | Bio-Rad Image Studio | https://www.licor.com/bio/products/Software, algorithm/image_studio/ | RRID: SCR_014210 | |
| Software, algorithm | Oroboros DatLab 6 V. 6.1.0.7 | https://www.oroboros.at/index.php/product/datlab/ | | |
| Other | Stainless Steel Beads 5 mm | Qiagen | Cat# 69989 | Lysis reagent |
| Other | Stuart Rotator Drive STR4 | Stuart-equipment | Cat# STR4 | Lysis reagent |
| Other | Hettich universal 320 R centrifuge | Andrea Hettich GmbH, Germany | Cat# 1406 | Lysis reagent |
| Other | $E_{max}$ precision microplate reader | Molecular Devices | Cat# LR88026 | Protein concentration quantification |
| Other | White 96-Well Immuno Plates | Fischer Scientific | Cat# 10415985 | Protein concentration quantification |
| Other | Fluoroskan, Microplate Fluorometer, one dispenser | Fischer Scientific | Cat# 5200111 | Protein concentration quantification |
| Other | Multiskan FC Microplate Photometer | Fischer Scientific | Cat# 11500695 | Protein concentration quantification |
| Other | PowerPac HC Power Supply | BioRad | Cat# 1645052 | Immunoblotting reagent |
| Other | TE 77 ECL Semi-Dry Transfer Unit | Amersham Biosciences | Cat# 80-6211-86 | Immunoblotting reagent |
| Other | ChemiDoc MP Imaging System | BioRad | Cat# 731BR00119 | Immunoblotting reagent |
| Other | 10% Criterion TGX Stain-Free gel | BioRad | Cat# 567–8035 | Immunoblotting reagent |
| Other | 16.5% Criterion Tris-Tricine/Peptide gel | BioRad | Cat# 345–0065 | Immunoblotting reagent |
| Other | Immobilon-P PVDF membrane | Millipore Sigma | Cat# IPVH00010 | Immunoblotting reagent |
| Other | Whatman cellulose chromatography papers | Millipore Sigma | Cat# WHA3030917 | Immunoblotting reagent |

| Reagent type (species) or resource | Designation | Source or reference | Identifiers | Additional information |
|---|---|---|---|---|
| Other | Sep-Pak C18 Cartrige | Waters | Cat# WAT054955 | Proteomic sample preparation |
| Other | Non-treated 96-Well Microplates | Fischer Scientific | Cat# 10252711 | Proteomic sample preparation |

