## [Editor Report]

The authors have measured the proteome and acetylome of human skeletal muscle before and after high intensity exercise. They found that many of the subunits of Complex V in mitochondria are selectively acetylated after exercise. This study will serve as a very useful resource for people interested in muscle and the acetyl proteome.

---

## [Decision Letter]

**Decision letter after peer review:**

Thank you for submitting your article "High-Intensity Interval Training Remodels the Proteome and Acetylome of Human Skeletal Muscle" for consideration by *eLife*. Your article has been reviewed by 3 peer reviewers, and the evaluation has been overseen by David James as the Senior and Reviewing Editor. The following individual involved in review of your submission has agreed to reveal their identity: Dr. Shinya Kuroda (Reviewer #1).

Essential revisions:

1) Further support for the conclusion that there is non-enzymatic acetylation is required. The effects of only two SIRT enzymes was reported and this needs to be expanded.

2) The highly descriptive nature of the paper was noted and while the data may serve as a useful resource, examples to illustrate the strength of this claim were not provided.

*Reviewer #1 (Recommendations for the authors):*

1. Although the authors suggest that increase in acetylation of mitochondrial proteins following HIIT would be via non-enzymatic mechanisms, there does not seem to be enough evidence for it. Addition of quantification of acetyl-CoA amounts in samples with enzymatic kit or mass spectrometry would strengthen the authors' conclusion. Otherwise, the claim of acetylation via non-enzymatic mechanisms should be toned down.

2. Figure 5E shows an increase in abundance of SIRT3 following HIIT. However, the increment looks small. I wonder whether such small increase consistently explain increase in acetylation of mitochondrial proteins. This issue would be further discussed. In addition, the abundance of acetyltransferase for mitochondrial proteins would be also discussed.

*Reviewer #2 (Recommendations for the authors):*

Strengths:

1) Very well-written manuscript with clear attention to detail.

2) The experimental methods employed are robust and the results were analyzed with well-thought-out and appropriate statistical procedures.

Weaknesses:

1) No post-hoc / orthogonal analyses were performed to assess the validity of the acetylome data.

2) The results are highly descriptive. The authors contend that the data contained within the manuscript could serve as "a substantial hypothesis-generating resource to stimulate further mechanistic research investigating how exercise improves metabolic health". While this may indeed be true, the authors have not provided any evidence to support their position. The manuscript would greatly benefit from a set of follow-up experiments which demonstrate how the data can lead to the discovery of new mechanisms.

*Reviewer #3 (Recommendations for the authors):*

1) The authors note that there are a substantial number of non-overlapping acetyl sites between the current study and that performed by Lundby et al. (2012). This is mentioned in the text, but there is no discussion of why this discrepancy might be apparent? If the same muscle was sampled, what might explain these discrepancies – different study population, different pulldown and mass spectrometry approach, variations in stringency of the analysis? A brief discussion of these (and other) possibilities would be useful for the non-expert.

2) Samples were taken 3 days after the last training bout, implying that the changes in acetylation are not necessarily a readout of the acute alterations in acetylation that might accompany the substantial changes in metabolite flux present during muscle contraction and the immediate recovery period. Some additional discussion around this would be valuable, particularly regarding the biological relevance of the identified acetylation changes at a timepoint that is temporally distinct from the time where many transcriptional and metabolic adaptations are directly induced by the exercise stimulus (i.e. in the immediate 24-48 hr after exercise). Furthermore, some discussion regarding the potential underlying molecular basis for the acetylation changes would benefit the manuscript. For example, are the changes in distinct acetylation sites the result of a newly 'reprogrammed' steady state of acetylation, or could they reflect sites that have become acetylated during exercise and simply have a slow turnover rate (i.e. deacetylation rate)?

3) The assessment of potential deacetylase enzymes was limited to two sirtuin proteins (SIRT1 and SIRT3), and while SIRT3 is the predominant deacetylase enzyme in the mitochondrial compartment, there a many other deacetylase enzymes that might play an important in regulating the acetylation of histone proteins (one of the novel findings of the current study). Assessment of an expanded range of deacetylase proteins could be considered, especially those outside of the mitochondrion.

4) If available, the authors should consider providing images to confirm equal protein loading for the immunoblots.

[Editors' note: further revisions were suggested prior to acceptance, as described below.]

Thank you for resubmitting your work entitled "High-Intensity Interval Training Remodels the Proteome and Acetylome of Human Skeletal Muscle" for further consideration by *eLife*. Your revised article has been evaluated by David James (Senior Editor) and a Reviewing Editor.

The manuscript has been improved but there are some remaining issues that need to be addressed, as outlined below:

As you can see, two of the reviewers remain concerned about the new experiment involving EP300 and would like to see some connectivity to exercise which is currently lacking. If you could provide at least your rationale for choosing this enzyme this might help placate the reviewers but ultimately, I think they are looking for a linker experiment and if you could provide this it would markedly uplift the study.

*Reviewer #1 (Recommendations for the authors):*

The authors correctly responded to my previous comments, and I am satisfied with their responses.

*Reviewer #2 (Recommendations for the authors):*

The authors have satisfactorily addressed several of the previous concerns that were raised but there are still a few issues that remain to be resolved.

1) It was previously suggested that the authors should provide some post-hoc / orthogonal analyses to assess the validity of the acetylome data. In response, western blots were performed to show that HIIT led to an increase in acetylation. The western blots provide support for the author's statement that "In general, there was a trend for increased acetylation following HIIT", but it does not provide an orthogonal means of validation for any of the specific sites that were concluded that have experienced a HIIT-induced change in acetylation. Could the authors please provide an orthogonal means of validation for at least one of the sites? Without such information, the reliability of the data remains questionable.

2) Previous comment: The authors contend that the data contained within the manuscript could serve as "a substantial hypothesis-generating resource to stimulate further mechanistic research investigating how exercise improves metabolic health". While this may indeed be true, the authors have not provided any evidence to support their position. The manuscript would greatly benefit from a set of follow-up experiments which demonstrate how the data can lead to the discovery of new mechanisms.

– In response to this comment, the authors performed siRNA-mediated knockdown of EP300 in C2C12 myotubes, and found the acetylation of H2B K20 to be reduced by the loss of EP300. The authors conclude that these experiments identify EP300 as an acetyltransferase regulating the acetylation of H2B K20 in skeletal muscle. The conclusion appears to be valid but it is not at all clear how the data reported in the study (i.e., the hypothesis-generating resource) led them to examine EP300. The authors need to explain how the data from their study led them to examine the EP300.

*Reviewer #3 (Recommendations for the authors):*

In the revised manuscript, the authors have addressed the majority of the concerns raised by the reviewers. I have a few follow up queries below that require attention.

The authors have expanded their analysis of enzymes that regulate protein acetylation status to potentially link any alterations in these enzymes to the altered acetylation levels observed in the current study. My concern with the additional analyses is the selection/measurement of SIRT5 as a deacetylase enzyme. There is a strong body of research that has shown that SIRT5 possesses very limited deacetylase activity, and instead has greater affinity for other acyl modifications (e.g. succinylation, malonylation) – PMID: 22076378, PMID: 21908771. On this basis it is unlikely that changes in SIRT5 would be directly involved in the exercise-induced changes in acetylation, and the importance of SIRT5 should be considered in this light.

A concern with this study is the highly descriptive nature of the experiments and the lack of verification of the importance of any of the described changes. To extend their findings, the authors have conducted a study in C2C12 myotubes assessing the impact of a specific deacetylase enzyme (EP300) on site-specific histone acetylation. The additional experiment has been performed in a robust manner and the experimental result is clear, namely that EP300 regulates H2B K20 acetylation. What is difficult to reconcile is how this experiment relates to the major findings of the manuscript? Acetylation marks can be potentially impacted by more than one enzyme, and as noted in this manuscript, there is substantial precedent in the literature for non-enzymatic acetylation to be a prominent mechanism regulating overall acetylation status. The authors present evidence that H2B K20 acetylation in human skeletal muscle is increased by exercise, but it is unclear if these changes in acetylation are in any way related to changes in the activity of EP300? Indeed, there is no evidence presented to directly link changes in EP300 abundance or activity with the reported exercise-induced changes in H2B K20 acetylation. As it stands, the new experiment identifies EP300 as a regulator of a specific acetylation site on H2B, but it does not provide any evidence for a causative role for EP300 in exercise-induced acetylation changes, and it also falls short of ascribing any functional relevance to the described changes in histone acetylation. Could the authors please comment.

[Editors' note: further revisions were suggested prior to acceptance, as described below.]

Thank you for resubmitting your work entitled "High-Intensity Interval Training Remodels the Proteome and Acetylome of Human Skeletal Muscle" for further consideration by *eLife*. Your revised article has been evaluated by David James (Senior Editor) and a Reviewing Editor.

The manuscript has been improved but there are some remaining issues that need to be addressed, as outlined below. As you can see Rev #3 is now satisfied that you have adequately revised the paper but Rev #2 is still of the opinion you could address two further comments. Could you please address these comments in full and resubmit your revised manuscript as soon as possible.

*Reviewer #2 (Recommendations for the authors):*

The effort put forth by the authors is appreciated but one of the primary concerns still remains.

1) It was previously suggested that the authors should provide some kind of post-hoc / orthogonal analysis to assess the validity of the acetylome data. In response, western blots were performed to show that HIIT led to an increase in acetylation. The western blots provide support for the author's statement that "In general, there was a trend for increased acetylation following HIIT", but it did not provide an orthogonal means of validation for any of the specific sites that were concluded that have experienced a HIIT-induced change in acetylation. The authors responded to this comment by indicating that there were limitations in the amount of remaining sample that they had, and they decided to prioritize the remaining sample for the "linker experiment". Understandably, there were limitations in the amount of sample that was available for post-hoc analyses, but unfortunately, this doesn't excuse the concerns with the validity of the outcomes. Without any kind of post-hoc / orthogonal validation, one is still left to wonder whether they can trust the site-specific omics data that was obtained.

*Reviewer #3 (Recommendations for the authors):*

The authors have clearly put a lot of time and effort into addressing my previous comments, and this is noted and appreciated. I have no further comments on the revised manuscript.

[Editors' note: further revisions were suggested prior to acceptance, as described below.]

Thank you for resubmitting your work entitled "High-Intensity Interval Training Remodels the Proteome and Acetylome of Human Skeletal Muscle" for further consideration by *eLife*. Your revised article has been evaluated by David James (Senior Editor) and a Reviewing Editor.

The manuscript has been improved but there are some remaining issues that need to be addressed, as outlined below:

We understand that in view of sample availability it is difficult to comply with all suggestions and comments.

In regards to Point 1 of Reviewer 2. I believe that the new data looking at one site should suffice.

In regards to Point 2, I think it will be important to address this sufficiently in the manuscript and potentially include the new figure for n=1 provided in your comments at least as a supplementary file and also discuss the point in the Results section of the manuscript.

---

## [Author Response]

Essential revisions:1) Further support for the conclusion that there is non-enzymatic acetylation is required. The effects of only two SIRT enzymes was reported and this needs to be expanded.

We have expanded upon this by measuring the effect of exercise training on two further sirtuins including those outside of the mitochondria (SIRT5 and SIRT6). As well as highlighting data for SIRT2, 3 and 5 from the proteome. In addition, we measured the abundance of the mitochondrial-localized acetyltransferase (GCN5L1) (see Figure 5E).

2) The highly descriptive nature of the paper was noted and while the data may serve as a useful resource, examples to illustrate the strength of this claim were not provided.

To test a hypothesis generated by these data, namely that acetylation of histones is likely controlled enzymatically via acetyltransferases such as EP300, we have performed knockdown studies in myotubes to investigate whether EP300 regulates the acetylation of specific acetyl-sites on histones that we found to be regulated by exercise. We identify H2B K20 as a EP300-regulated acetyl-site (see Figure 7E).

Reviewer #1 (Recommendations for the authors):1. Although the authors suggest that increase in acetylation of mitochondrial proteins following HIIT would be via non-enzymatic mechanisms, there does not seem to be enough evidence for it. Addition of quantification of acetyl-CoA amounts in samples with enzymatic kit or mass spectrometry would strengthen the authors' conclusion. Otherwise, the claim of acetylation via non-enzymatic mechanisms should be toned down.

Unfortunately, due to the scarcity of sample that can be collected during a human skeletal muscle biopsy using the Bergström needle technique, we do not have sufficient sample remaining to measure acetyl-CoA. However, we refer you to lines 360 to 364, where we discuss the available literature regarding the effect of acute and chronic exercise on acetyl-CoA concentration in skeletal muscle. Nonetheless, we accept your concern that our conclusions relating to non-enzymatic mechanisms may be too strong. As such, we have toned this down by removing “likely by non-enzymatic mechanisms” from the abstract (line 31), and removing the sentence “Altogether, the data contained within Figure 5 indicate a predominantly non-enzymatic acetylation of mitochondrial proteins during HIIT.” from the discussion on lines 390-391.

2. Figure 5E shows an increase in abundance of SIRT3 following HIIT. However, the increment looks small. I wonder whether such small increase consistently explain increase in acetylation of mitochondrial proteins. This issue would be further discussed. In addition, the abundance of acetyltransferase for mitochondrial proteins would be also discussed.

SIRT3 protein abundance increases by approximately 60%, as measured via immunoblotting. In the context of exercise training, we interpret this to be a robust increase. We have measured the abundance of the mitochondrial acetyltransferase GCN5L1 via immuoblotting and find that this is unchanged in response to exercise training (see Figure 5E). This is reported on lines 383-385.

Reviewer #2 (Recommendations for the authors):Strengths:1) Very well-written manuscript with clear attention to detail.2) The experimental methods employed are robust and the results were analyzed with well-thought-out and appropriate statistical procedures.

Thank you for your positive responses.

Weaknesses:1) No post-hoc / orthogonal analyses were performed to assess the validity of the acetylome data.

We have now included an immunoblot for pan-acetylation within skeletal muscle, which confirms the predominant increase in acetylation following exercise training (see Figure 5—figure supplement 1).

2) The results are highly descriptive. The authors contend that the data contained within the manuscript could serve as "a substantial hypothesis-generating resource to stimulate further mechanistic research investigating how exercise improves metabolic health". While this may indeed be true, the authors have not provided any evidence to support their position. The manuscript would greatly benefit from a set of follow-up experiments which demonstrate how the data can lead to the discovery of new mechanisms.

To identify the mechanism by which the exercise-responsive histone acetylation sites may be regulated we investigated the role of the acetyltransferase EP300. Using siRNA-mediated knockdown of EP300 in C2C12 myotubes, we found the acetylation of H2B K20 to be reduced with the loss of EP300 (Figure 7E). These experiments identify EP300 as an acetyltransferase regulating the acetylation of H2B K20 in skeletal muscle. Please see lines 476-481.

Reviewer #3 (Recommendations for the authors):1) The authors note that there are a substantial number of non-overlapping acetyl sites between the current study and that performed by Lundby et al. (2012). This is mentioned in the text, but there is no discussion of why this discrepancy might be apparent? If the same muscle was sampled, what might explain these discrepancies – different study population, different pulldown and mass spectrometry approach, variations in stringency of the analysis? A brief discussion of these (and other) possibilities would be useful for the non-expert.

We have included a brief discussion of this on lines 227-230. We believe that this discrepancy is due to both the induction of acetylation by exercise revealing previously undetectable acetylation sites, stochastic nature of data dependent acquisition methods, differences in LC-MS/MS instruments, as well as updated FASTA files.

2) Samples were taken 3 days after the last training bout, implying that the changes in acetylation are not necessarily a readout of the acute alterations in acetylation that might accompany the substantial changes in metabolite flux present during muscle contraction and the immediate recovery period. Some additional discussion around this would be valuable, particularly regarding the biological relevance of the identified acetylation changes at a timepoint that is temporally distinct from the time where many transcriptional and metabolic adaptations are directly induced by the exercise stimulus (i.e. in the immediate 24-48 hr after exercise). Furthermore, some discussion regarding the potential underlying molecular basis for the acetylation changes would benefit the manuscript. For example, are the changes in distinct acetylation sites the result of a newly 'reprogrammed' steady state of acetylation, or could they reflect sites that have become acetylated during exercise and simply have a slow turnover rate (i.e. deacetylation rate)?

It is our belief that this timepoint is not temporally distinct from changes in metabolism, demonstrated by the increased mitochondrial respiratory capacity in the same muscle biopsies (i.e. also 3 days after the last training bout). We have expanded our discussion of the possible relationship between acetylation of mitochondrial proteins and mitochondrial respiratory capacity and fat oxidation on lines 428-446.

It is our belief that the increased acetylation after exercise training is a steady-state and that it may occur from accumulation of protein copies/sites becoming acetylated during exercise, when acetyl-coA levels are elevated, which are then either not deacetylated or not at a rate that is sufficient to reduce acetylation prior to the next exercise-induced acetylation event. However, this is purely speculative and, indeed, work in rodents may suggest otherwise (see Overmyer et al., 2015), so we are reticent about over-speculation in the discussion. We do discuss these possibilities and how they relate to acetyl-coA fluxes and rates of fatty acid metabolism on lines 356-368.

3) The assessment of potential deacetylase enzymes was limited to two sirtuin proteins (SIRT1 and SIRT3), and while SIRT3 is the predominant deacetylase enzyme in the mitochondrial compartment, there a many other deacetylase enzymes that might play an important in regulating the acetylation of histone proteins (one of the novel findings of the current study). Assessment of an expanded range of deacetylase proteins could be considered, especially those outside of the mitochondrion.

We have extended our examination of deacetylase abundance to include SIRT1, 3, 5 and 6. In addition we have assessed the abundance of the mitochondrial acetyltransferase GCN5L1. Please see Figure 5E and the results/discussion on lines 372-391.

4) If available, the authors should consider providing images to confirm equal protein loading for the immunoblots.

Representative “loading control” images have been provided in Figure 5—figure supplement 2, and reference made between corresponding immunoblots and loading controls in each figure legend.

[Editors' note: further revisions were suggested prior to acceptance, as described below.]

The manuscript has been improved but there are some remaining issues that need to be addressed, as outlined below:As you can see, two of the reviewers remain concerned about the new experiment involving EP300 and would like to see some connectivity to exercise which is currently lacking. If you could provide at least your rationale for choosing this enzyme this might help placate the reviewers but ultimately, I think they are looking for a linker experiment and if you could provide this it would markedly uplift the study.Reviewer #2 (Recommendations for the authors):The authors have satisfactorily addressed several of the previous concerns that were raised but there are still a few issues that remain to be resolved.1) It was previously suggested that the authors should provide some post-hoc / orthogonal analyses to assess the validity of the acetylome data. In response, western blots were performed to show that HIIT led to an increase in acetylation. The western blots provide support for the author's statement that "In general, there was a trend for increased acetylation following HIIT", but it does not provide an orthogonal means of validation for any of the specific sites that were concluded that have experienced a HIIT-induced change in acetylation. Could the authors please provide an orthogonal means of validation for at least one of the sites? Without such information, the reliability of the data remains questionable.

Owing to the scarce amount of material remaining (due to the limited specimen sizes from human skeletal muscle biopsies), and our decision to prioritise providing a “linker experiment” between our exercise acetylome data and the EP300 knockdown experiments (and, in doing so, addressing your comment #2 and reviewer 3’s similar concerns) we were left without sufficient material to perform additional immunoblots for site- and protein-specific acetylation. We hope that you understand the difficulty of prioritising analyses when specimen amount is limited.

2) Previous comment: The authors contend that the data contained within the manuscript could serve as "a substantial hypothesis-generating resource to stimulate further mechanistic research investigating how exercise improves metabolic health". While this may indeed be true, the authors have not provided any evidence to support their position. The manuscript would greatly benefit from a set of follow-up experiments which demonstrate how the data can lead to the discovery of new mechanisms.– In response to this comment, the authors performed siRNA-mediated knockdown of EP300 in C2C12 myotubes, and found the acetylation of H2B K20 to be reduced by the loss of EP300. The authors conclude that these experiments identify EP300 as an acetyltransferase regulating the acetylation of H2B K20 in skeletal muscle. The conclusion appears to be valid but it is not at all clear how the data reported in the study (i.e., the hypothesis-generating resource) led them to examine EP300. The authors need to explain how the data from their study led them to examine the EP300.

We thank you for this comment. We were led to examine EP300 by our observations of elevated histone acetylation following HIIT, alongside our previous observation that EP300 is an acetyltransferase targeting a wide-range of histone acetyl-sites in MEF and kasumi-1 cells (Weinert et al., 2018), however we neglected to explain this, nor cite this article, which we have now rectified (lines 456-458). In addition, we have now measured EP300 in the human skeletal muscle biopsies before and after HIIT – identifying a upregulation following training (after the exclusion of an outlier). Please see lines 458-460, figure 7E, and lines 877-881 (the legend for Figure 7). Together, we believe that these data provide a solid rationale for investigating EP300 as an acetyltransferase regulating the acetylation of H2B K20 in skeletal muscle.

Reviewer #3 (Recommendations for the authors):In the revised manuscript, the authors have addressed the majority of the concerns raised by the reviewers. I have a few follow up queries below that require attention.The authors have expanded their analysis of enzymes that regulate protein acetylation status to potentially link any alterations in these enzymes to the altered acetylation levels observed in the current study. My concern with the additional analyses is the selection/measurement of SIRT5 as a deacetylase enzyme. There is a strong body of research that has shown that SIRT5 possesses very limited deacetylase activity, and instead has greater affinity for other acyl modifications (e.g. succinylation, malonylation) – PMID: 22076378, PMID: 21908771. On this basis it is unlikely that changes in SIRT5 would be directly involved in the exercise-induced changes in acetylation, and the importance of SIRT5 should be considered in this light.

Thank you for highlighting this point. We have modified the text on lines 368-372 to reflect this.

A concern with this study is the highly descriptive nature of the experiments and the lack of verification of the importance of any of the described changes. To extend their findings, the authors have conducted a study in C2C12 myotubes assessing the impact of a specific deacetylase enzyme (EP300) on site-specific histone acetylation. The additional experiment has been performed in a robust manner and the experimental result is clear, namely that EP300 regulates H2B K20 acetylation. What is difficult to reconcile is how this experiment relates to the major findings of the manuscript? Acetylation marks can be potentially impacted by more than one enzyme, and as noted in this manuscript, there is substantial precedent in the literature for non-enzymatic acetylation to be a prominent mechanism regulating overall acetylation status. The authors present evidence that H2B K20 acetylation in human skeletal muscle is increased by exercise, but it is unclear if these changes in acetylation are in any way related to changes in the activity of EP300? Indeed, there is no evidence presented to directly link changes in EP300 abundance or activity with the reported exercise-induced changes in H2B K20 acetylation. As it stands, the new experiment identifies EP300 as a regulator of a specific acetylation site on H2B, but it does not provide any evidence for a causative role for EP300 in exercise-induced acetylation changes, and it also falls short of ascribing any functional relevance to the described changes in histone acetylation. Could the authors please comment.

To address the link between EP300, H2B K20 acetylation, and exercise we have performed several followup experiments. Firstly, we assessed whether EP300 is regulated by HIIT in human skeletal muscle. Via immunoblotting, we were able to identify the upregulation of EP300 following training (after the exclusion of an outlier). Please see lines 458-460, figure 7E, and lines 877-881. This finding is in line with the changes in H2B K20 acetylation following HIIT (figure 7B), and the in vitro regulation of H2B K20 by EP300 in C2C12 myotubes (figure 7F). We believe that, collectively, these data provide evidence for a role of EP300 in regulating H2B K20 in skeletal muscle, and that EP300 and H2B K20 acetylation are coregulated during exercise training.

Nonetheless, we agree with you that this stops short of a truly causal relationship and we have attempted to establish a model in which we could investigate causality. Although mimicking exercise in vitro is challenging, we performed electrical pulse stimulation (EPS) contraction experiments in C2C12 myotubes. We used a 24-h contraction protocol, which we have previously used to study exercise responses (Gonzalez-Franquesa et al., PMID: 34495765), with the long duration of contraction designed to mimic a training response. Unfortunately, this approach did not alter pan-acetylation nor the acetylation of H2B K20 (see Figure below). These observations rendered this model unsuitable for the direct assessment of EP300 in contraction/exercise-induced H2B K20 acetylation. To truly establish a causal link between EP300 and H2B K20 acetylation in response to exercise, we believe that in vivo studies using p300 knockdown models would be required. However, given the substantial compensation/redundancy between p300 and CBP in vivo (PMIDs 26712218, 30888860) and the lethality of skeletal muscle specific double CBP/p300 knockout mice (PMID: 31898871) these experiments would be highly challenging, comprising a whole research project/manuscript in their own right. Thus, these experiments are outside the scope of this investigation.

We hope that you appreciate our efforts in addressing your concern and recognise that by identifying the upregulation of EP300 following HIIT, we have provided an additional link between exercise-induced acetylation of H2B K20 and activation of EP300.

**Author response image 1. sa2fig1:** No change in (A) pan-lysine acetylation, or (B) H2B K20 acetylation following 24 hours of electrical pulse stimulation in C2C12 myotubes (n = 6/ group).

[Editors' note: further revisions were suggested prior to acceptance, as described below.]

The manuscript has been improved but there are some remaining issues that need to be addressed, as outlined below.Reviewer #2 (Recommendations for the authors):The effort put forth by the authors is appreciated but one of the primary concerns still remains.1) It was previously suggested that the authors should provide some kind of post-hoc / orthogonal analysis to assess the validity of the acetylome data. In response, western blots were performed to show that HIIT led to an increase in acetylation. The western blots provide support for the author's statement that "In general, there was a trend for increased acetylation following HIIT", but it did not provide an orthogonal means of validation for any of the specific sites that were concluded that have experienced a HIIT-induced change in acetylation. The authors responded to this comment by indicating that there were limitations in the amount of remaining sample that they had, and they decided to prioritize the remaining sample for the "linker experiment". Understandably, there were limitations in the amount of sample that was available for post-hoc analyses, but unfortunately, this doesn't excuse the concerns with the validity of the outcomes. Without any kind of post-hoc / orthogonal validation, one is still left to wonder whether they can trust the site-specific omics data that was obtained.

We have now provided additional immunoblots against specific acetyl-sites on SOD2 (K68 and K122; Figure 5 – supplemental figure 1B), which display increases in acetylation that reflects the changes seen in the acetylome (Figure 5A). Additionally, in the absence of further sample, we attempted to strip the membranes and re-probe for total SOD2, however the signal did not persist after stripping. Nonetheless it is our opinion that the acetyl-blots *per se* are the correct validation for the changes in acetyl-peptide abundances presented in the acetylome.

[Editors' note: further revisions were suggested prior to acceptance, as described below.]

The manuscript has been improved but there are some remaining issues that need to be addressed, as outlined below:We understand that in view of sample availability it is difficult to comply with all suggestions and comments. In regards to Point 1 of Reviewer 2. I believe that the new data looking at one site should suffice. In regards to Point 2, I think it will be important to address this sufficiently in the manuscript and potentially include the new figure for n=1 provided in your comments at least as a supplementary file and also discuss the point in the Results section of the manuscript.

We thank you for your support of this manuscript and your understanding regarding sample availability. To address your remaining concerns, we have included a comparison of the normalized and non-normalized EP300 immunoblot data as Figure7—figure supplement 1, as well as discussing this in text (lines 460-465).